# FROM SINGLE TO MULTI-GRANULARITY: TOWARD LONG-TERM MEMORY ASSOCIATION AND SELECTION OF CONVERSATIONAL AGENTS

**Derong Xu**[1,2]**, Yi Wen**[2]**, Pengyue Jia**[2]**, Yingyi Zhang**[2,4]**, Wenlin Zhang**[2]**, Yichao Wang**[3,*]**,
Huifeng Guo**[3]**, Ruiming Tang**[3]**, Xiangyu Zhao**[2,*]**, Enhong Chen**[1]**, Tong Xu**[1,*]
[1]University of Science and Technology of China,[2]City University of Hong Kong,
[3]Huawei Technologies Ltd., [4]Dalian University of Technology
`derongxu@mail.ustc.edu.cn,xianzhao@cityu.edu.hk,`
`wangyichao5@huawei.com,tongxu@ustc.edu.cn`

## ABSTRACT

Large Language Models (LLMs) have recently been widely adopted in conversational agents. However, the increasingly long interactions between users and agents accumulate extensive dialogue records, making it difficult for LLMs with limited context windows to maintain a coherent long-term dialogue memory and deliver personalized responses. While retrieval-augmented memory systems have emerged to address this issue, existing methods often depend on single-granularity memory segmentation and retrieval. This approach falls short in capturing deep memory connections, leading to partial retrieval of useful information or substantial noise, resulting in suboptimal performance. To tackle these limits, we propose MemGAS, a framework that enhances memory consolidation by constructing multi-granularity association, adaptive selection, and retrieval. MemGAS is based on multi-granularity memory units and employs Gaussian Mixture Models to cluster and associate new memories with historical ones. An entropy-based router adaptively selects optimal granularity by evaluating query relevance distributions and balancing information completeness and noise. Retrieved memories are further refined via LLM-based filtering. Experiments on four long-term memory benchmarks demonstrate that MemGAS outperforms state-of-the-art methods on both question answer and retrieval tasks, achieving superior performance across different query types and top-K settings[1].

## 1 INTRODUCTION

Large Language Models (LLMs) have showcased remarkable conversational abilities, enabling them to serve as personalized assistants (Li et al., 2025a; Wang et al., 2024; Li et al., 2024b; Zhang et al., 2025b) and handle a wide range of applications (Gao et al., 2024; Wang et al., 2023; Fu et al., 2025), such as customer service (Pandya & Holia, 2023) and software development (Qian et al., 2024). However, as user-agent interactions increase, the volume of conversational history grows significantly. Despite their impressive conversational abilities, LLMs struggle with maintaining long-term conversational memory due to their limited context length (Liu et al., 2023b), which makes it challenging to retain a comprehensive record of user interactions and preferences over time. This limitation undermines their ability to generate coherent and personalized responses (Deng et al., 2026; Wu et al., 2025a; Pan et al., 2025; Xu et al., 2025b; Li et al., 2025c). The development of retrieval-based external (non-parametric) memory systems (Packer et al., 2023; Park et al., 2023) has emerged as a promising solution. By storing interaction histories and retrieving relevant information when needed, memory systems enable LLMs to recall user-specific details from past dialogues and deliver more tailored responses.

---

[*]Corresponding authors.
[1]https://github.com/Applied-Machine-Learning-Lab/ICLR2026_MemGAS

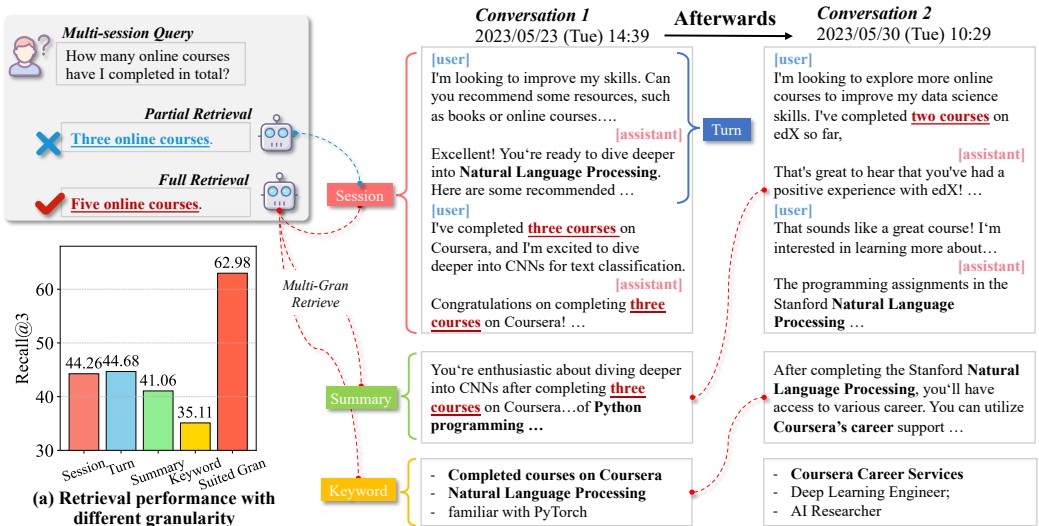

Figure 1: An example of the multi-session query. The agent needs to synthesize information from both *Conversation 1* and *Conversation 2* to answer the question. The **red underlines** highlight information directly relevant to the answer, while the **bold text** emphasizes the same details between conversations. By generating multi-granularities (e.g., summary and keyword), the agent can better establish connections between conversations, enhancing full memory retrieval. Additionally, chart (a) illustrates the performance across different granularities, where 'Suited Gran' means the result using best-suited granularity for each query.

Recent studies on external memory systems of LLM agents primarily depend on the Retrieval-Augmented Generation (RAG) pipeline and have explored diverse aspects of memory segmentation and construction for effective retrieval (Jia et al., 2025; Xu et al., 2025a; Zhang et al., 2025c; Wu et al., 2025b; Zhang et al., 2026; 2024a; Zhong et al., 2024; Li et al., 2025b). For memory segmentation, existing approaches mainly adopt single-granularity to segment conversations. They utilize session-level chunks as retrieval units (Lu et al., 2023; Liu et al., 2023a; Team, 2023), while others employ finer-grained turn-level segmentation to capture details (Zhong et al., 2024; Wu et al., 2025a; Yuan et al., 2023). Recent advances introduce topic-aware segmentation techniques (Pan et al., 2025; Tan et al., 2025) that group dialogue based on semantic coherence, enhancing topic-consistent retrieval. Additionally, several works (Wang et al., 2025; Team, 2023; Zhong et al., 2024; Sarthi et al., 2024; Chen et al., 2025a) generate memory summaries, condensing key information into compact representations to improve retrieval efficiency. Regarding memory construction, researchers have explored structured organization paradigms to enhance long-term knowledge retention (Li et al., 2024c; Edge et al., 2024; Guo et al., 2024). Methods like RAPTOR (Sarthi et al., 2024) and MemTree (Rezazadeh et al., 2024) employ tree structures to encode multi-scale relations between memory units. Some works propose hierarchical memory architectures (Sun & Zeng, 2025; Hu et al., 2025), building complex systems with deep abstraction capabilities that enable efficient top-down retrieval. Other studies (Gutiérrez et al.; 2025; Chhikara et al., 2025; Rasmussen et al., 2025) implement graph-based memory architectures to simulate neural memory consolidation processes, explicitly modeling relational structures between entities.

However, despite their progress, current approaches exhibit two critical limits: **i) Insufficient Multi-Granularity Memory Connection.** While existing methods endeavor to organize memories into topological structures (e.g., knowledge graphs (Gutiérrez et al., 2025; Chhikara et al., 2025) or trees (Sarthi et al., 2024; Rezazadeh et al., 2024)), they predominantly concentrate on singular granularity levels—either entities or session summaries. This single-scale paradigm fails to model cross-granular interactions between memory units, resulting in retrieving only partial useful information. As demonstrated in Figure 1, answering multi-session queries requires establishing semantic links across granularities (e.g., connecting *Conversation 1* and *Conversation 2* through shared keyword/summary). Failure to establish links results in partial retrieval (e.g., retrieving only *Conversation 1*), which leads to incorrect answers. **ii) Lack of Adaptive Multi-Granular Memory Selection.** Current

methods mainly rely on fixed granularity strategies (e.g., session/turn segmentation or LLM-generated summaries (Wang et al., 2025; Zhong et al., 2024; Wu et al., 2025a)), which often lead to incomplete context recall or noise due to improper granularity. Although topic-aware segmentation (Pan et al., 2025; Tan et al., 2025) enhances intra-chunk coherence, they lack adaptive mechanisms to select granularity for each query. Our empirical analysis in Figure 1(a) reveals that adaptively choosing best-suited granularity per query (e.g., balancing noise reduction in summaries/keywords with information retention in raw sessions) yields substantial performance gains. This highlights the necessity of granularity selection to resolve the inherent noise-information trade-off.

In this paper, we propose MemGAS, a framework for constructing and retrieving long-term memories through multi-granularity association and adaptive selection. Our method addresses these issues by two core strategies: **i) Memory Association:** We leverage LLMs to generate memory summaries and keywords, constructing multi-granular memory units. When new memory is updated, a Gaussian Mixture Model is employed to cluster historical memories into an accept set (relevant) and a reject set (irrelevant). The memories in the accept set are then associated with new memories, ensuring consolidated memory structures and real-time updates. **ii) Granularity Selection:** An entropy-based router adaptively assigns retrieval weights to different granularities by evaluating the certainty of the query's relevance distribution. Finally, critical memories are retrieved using Personalized PageRank and filtered through LLMs to remove redundancies, ensuring a refined and high-quality memory that enhances the assistant's understanding. Experiments on four open-source long-term memory benchmarks demonstrate that MemGAS significantly outperforms state-of-the-art baselines and single-granularity approaches on both question answering (QA) and retrieval tasks. Moreover, it consistently achieves superior results across various query types and retrieval top-k settings.

## 2 METHODOLOGY

This section defines the task and data format, constructs a dynamical memory association framework, details an entropy-based router for better-suited granularity, and outlines strategies for retrieving and filtering high-quality contextual information for response generation.

### 2.1 PRELIMINARY

Our work focuses on building personalized assistants through long-term conversational memory, where the system leverages multi-session user-agent interactions (referred to as memory) to construct an external memory bank $\mathcal{M}$. Without loss of generality, the $i$-th session $S_i = \{(u_j^{(i)}, a_j^{(i)})\}_{j=1}^{n_i}$ contains $n_i$ dialogue turns of user utterances $u_j^{(i)}$ and assistant responses $a_j^{(i)}$. When the assistant receives a query $q$, our aim is to retrieve relevant memories $\mathcal{M}_{\text{rel}} \subseteq \mathcal{M}$ through multi-granular associations across session-level $S$, turn-level $T$, keyword-level $K$, and summary-level $U$, then generates responses via $a = \text{LLM}(q, \mathcal{M}_{\text{rel}})$. The core challenges involve establishing cross-granular memory association and learning adaptive granularity selection $\psi : q \to \{\alpha_s, \alpha_t, \alpha_k, \alpha_u\}$ to balance information completeness and retrieval noise.

### 2.2 MULTI-GRANULARITY ASSOCIATION CONSTRUCTION

Existing methods mainly encode memories into vector libraries (e.g., session-level chunks) and directly retrieve information via similarity search (Izacard et al., 2021; Lee et al., 2023; Lu et al., 2023). However, such approaches overlook deeper associations between memories. To address this, we propose an associative memory construction process that captures multi-granular relationships.

**Multi-Granular Memory Metadata.** For the $i$-th session $S_i$, multi-granularity metadata is generated using LLMs, including a session summary $U_i$ and keywords $K_i$. Additionally, the session is segmented into multiple turns $T_i$. Formally,

$$U_i, K_i = f_{\text{LLM}}(S_i), \quad T_i = \text{segment}(S_i) \tag{1}$$

Here, $f_{\text{LLM}}$ denotes LLM processing function, and segment represents segmentation operation. The resulting memory chunk $M_i$ combines the session-level $S_i$, turn-level dialogues $T_i$, summary $U_i$, and keywords $K_i$, and is stored in memory bank as:

$$M_i = \{S_i, T_i, U_i, K_i\} \in \mathcal{M} \tag{2}$$

**(1) Multi-Granularity Association Construction**

Figure 2: Overall Framework. In the first phase, we leverage LLM agent to generate multi-granularity information and construct Memory Association using GMM. In the second phase, given a query, we perform multi-granularity routing based on entropy and utilize PPR algorithm to search for key nodes. Finally, we filter the information to obtain the answer.

**Dynamical Memory Association.** When a new memory $\mathcal{M}_{\text{new}}$ is added, the current memory bank $\mathcal{M}_{\text{cur}}$ is updated as $\mathcal{M}_{\text{cur}} \leftarrow \mathcal{M}_{\text{cur}} \cup \mathcal{M}_{\text{new}}$. To establish associations between $\mathcal{M}_{\text{new}}$ and historical memories, a Gaussian Mixture Model (GMM)-based clustering strategy (Rasmussen, 1999) is used. Each granularity of each element in the memory bank is encoded as a dense vector. Specifically, all memories are encoded into dense vectors $e(S_i)$, $e(T_i)$, $e(U_i)$, and $e(K_i)$ for session-level, turn-level, summary, and keyword metadata, respectively. The entire memory bank $\mathcal{M}_{\text{cur}}$ is thus represented as a collection of these multi-granular vectors. Pairwise similarity scores are computed between $e(\mathcal{M}_{\text{new}})$ and $e(\mathcal{M}_{\text{cur}})$, covering all granularities of each memory. The resulting set of similarity vectors $\mathbf{s}_{\text{sim}}$ is clustered by GMM into two probabilistic sets:

- **Accept Set**: Memories with high similarity to $\mathcal{M}_{\text{new}}$, forming direct associations with $\mathcal{M}_{\text{new}}$.

- **Reject Set**: Irrelevant memories, excluded from immediate connections with $\mathcal{M}_{\text{new}}$.

Note that the similarity vectors are computed with granularity-specific information, meaning that each granularity of $\mathcal{M}_{\text{new}}$ is treated as a node and used to establish connections with nodes in $\mathcal{M}_{\text{cur}}$. We maintain an association graph $\mathcal{A}_{\text{cur}}$ to store the connections in $\mathcal{M}$, which is updated as $\mathcal{A}_{\text{cur}} \leftarrow \mathcal{A}_{\text{cur}} \cup \mathcal{A}_{\text{new}}$, where $\mathcal{A}_{\text{new}}$ represents the edges between $\mathcal{M}_{\text{new}}$ and its **accept set**. This process mimics human-like memory consolidation by selectively reinforcing contextually related memories.

## 2.3 MULTI-GRANULARITY ROUTER

Existing methods rely on single predefined granularity for memory retrieval, limiting their ability to adaptively prioritize fine- or coarse-grained information based on query (Lee et al., 2023; Wang et al., 2025; Sarthi et al., 2024). To address this, we propose an entropy-based router that adaptively selects the better-suited granularity for each query.

**Entropy-Driven Granularity Selection.** For a query $q$, we first compute its similarity to all memory chunks across each granularity level $g \in \{\text{session, turn, summary, keyword}\}$. Let $\mathbf{s}^g = [\text{sim}(q, M_1^g), \ldots, \text{sim}(q, M_n^g)]$ denote the similarity scores between $q$ and n memories chunks at granularity $g$. We normalize $\mathbf{s}^g$ into a probability distribution $p^g(\mathbf{s}^g)$ and calculate its Shannon entropy:

$$H^g = -\sum_{i=1}^{n} p_i^g(\mathbf{s}^g) \log p_i^g(\mathbf{s}^g), \text{ where } p_i^g(\mathbf{s}^g) = \frac{\exp(\text{sim}(q, M_i^g)/\lambda)}{\sum_{j=1}^{n} \exp(\text{sim}(q, M_j^g)/\lambda)}, \forall i \in \{1, \ldots, n\}.$$

(3)

where $H^g$ quantifies the uncertainty of matching $q$ to memories at granularity $g$. $\lambda$ is a hyperparameter to control the degree of entropy, and is analyzed in Appendix F.

**Soft Router Weights.** Our motivation stems from that lower entropy $H^g$ typically reflects higher confidence in precise matches (e.g., higher confidence indicates a clear correspondence between the query and its associated memory). Conversely, higher entropy suggests more ambiguous matches (e.g., lower confidence implies uncertainty about the query's corresponding memory). To capture this behavior, we assign weights to granularities by normalizing their inverse entropy:

$$w^g = \frac{1/H^g}{\sum_{g'=1}^{G} 1/H^{g'}}, \tag{4}$$

where $G$ denotes the total number of granularities. This formulation ensures that granularities with lower entropy (indicating higher certainty) are assigned greater weights. Consequently, memories are reweighted to emphasize those associated with the query's most confident granularities, eliminating the need for manual intervention. We present a theoretical analysis of granularity association and routers in Appendix H.

## 2.4 MEMORY RETRIEVAL AND FILTER

After constructing the multi-granularity memory associations and determining granularity weights, we retrieve relevant memories for a query $q$ by leveraging the graph-structured memory $\{M_i, A_i\} \in \mathcal{M} \times \mathcal{A}$. To fully utilize inter-memory relationships, we employ the Personalized PageRank (PPR) algorithm (Haveliwala, 2002) for context-aware ranking.

At the granularity level, we treat each $M_i^g$ (i.e., the $g$-th granularity node of memory $M_i$) as an individual node in the association graph. For each granularity $g$, we compute an initial relevance score for node $M_i^g$ using the router-assigned weight $w^g$ from Equation 4:

$$\text{score}_i^g = w^g \cdot \text{sim}(q, M_i^g), \tag{5}$$

where $\text{sim}(q, M_i^g)$ measures the cosine similarity between the query embedding $e(q)$ and the granularity-specific embedding $e(M_i^g)$. The set of scores $\{\text{score}_i^g\}$ defines the personalized starting probabilities over granularity-level nodes. We then select the top $\alpha$ nodes as seed nodes (analyzed in Appendix F), and run PPR on the multi-granularity association graph, propagating relevance through the graph structure to emphasize nodes that are both directly relevant to the query and densely connected to other high-value nodes. After convergence, we select the top-$k$ nodes as candidate contexts by their final PPR scores. The impact of $K$ is empirically analyzed in Section § 3.4.

**LLM-Based Redundancy Filtering.** To minimize noise and eliminate redundancy in the retrieved multi-granularity memories, we employ an LLM-based filtering mechanism. This mechanism processes the top-$K$ memories alongside the query $q$, using a curated designed prompt (see Appendix J) to identify and discard irrelevant or repetitive content. As a result, the final context provided to the response generator is refined to focus exclusively on the most critical information relevant to $q$, ensuring a more concise and personalized response. We present case study on multi-granularity information in Appendix K.2 and case study on filtering in Appendix K.3.

## 3 EXPERIMENTS

### 3.1 EXPERIMENTAL SETTINGS

**Dataset and Metrics.** Experiments are conducted on four comprehensive long-term memory datasets: LoCoMo (Maharana et al., 2024), Long-MT-Bench+ (Pan et al., 2025), LongMemEval-s (Wu et al., 2025a), and LongMemEval-m (Wu et al., 2025a), all focused on evaluating the capabilities of LLM agents in long-term conversations. Since our task is training-free, the whole QA pairs in datasets are used for evaluation. Detailed dataset statistics are provided in Appendix A. To comprehensively evaluate model performance, we employ multiple metrics: F1 scores (as used by Maharana et al. (2024)), BLEU (with 4-gram by default), BERTScore, and ROUGE scores (following Pan et al. (2025)). Additionally, we introduce GPT4o-as-Judge (GPT4o-J) (Zheng et al., 2023), an evaluation setting where GPT4o assesses the alignment of a model's response with the reference answer. The evaluation prompts are provided in Appendix J.

**Baselines**. We compare MemGAS against various methods. (1) **Full History**: which utilizes all the latest conversation records, incorporating up to 128k tokens of context. *Two strong retrieval models*: (2) **MPNet** (Song et al., 2020) and (3) **Contriever** (Izacard et al., 2021). *Four memory-based conversational models*: (4) **RecurSum** (Wang et al., 2025), which uses LLMs to recursively summarize and update memory for contextually relevant responses; (5) **MPC** (Lee et al., 2023), which composes LLM with prompting and external memory; (6) **A-Mem** (Xu et al., 2025b), which organizes memories through generating note and links; (7) **SeCom** (Pan et al., 2025), which segments memory into coherent topics and applies compression-based denoising to boost retrieval; *Two structured RAG models*: (8) **HippoRAG 2** (Gutiérrez et al.), which integrates knowledge graphs for efficient retrieval, and (9) **RAPTOR** (Sarthi et al., 2024), which enhances retrieval via recursive summary and hierarchical clustering into a tree structure. We also include two recent memory models: H-MEM (Sun & Zeng, 2025) introduces a four-layer hierarchical memory with positional indexing, and COMEDY Chen et al. (2025a) provides a unified One-for-All compressive memory framework without traditional memory stores. Due to space limitations, the experimental results for H-MEM and COMEDY are provided in Appendix I.2. More details can be found in Appendix B.

**Implementation Details.** We use 'gpt-4o-mini-2024-07-18' as the backbone for all tasks, including multi-granularity information generation and QA. To ensure fairness, all baselines share consistent generation prompts. The temperature of LLMs is set to 0 for reproducibility, and all models operate in a zero-shot setting with prompt templates detailed in Appendix J. A consistent top-3 session retrieval setting is applied across all models for fair comparison, with top-3 segments used for SeCom and sessions/summaries for RAPTOR. Contriever is used as the encoding model to generate embeddings for memory texts. We select hyperparameters through grid search: $\lambda$ is tuned over {0.1, 0.2, 0.3, 0.5, 0.7, 1.0} and $\alpha$ over {5, 10, 15, 20, 25}. LongMTBench+ is excluded due to the lack of a ground truth for retrieval. RAPTOR and A-Mem retrieval cannot be evaluated with session-level Recall because their stored units are abstracted or rewritten representations, so there is no deterministic mapping back to ground-truth sessions. Results for RAPTOR, A-Mem, and HippoRAG on LongMemEval-m are unavailable due to high runtime.

We compared various methods across different retrievers, generators, and query types, as detailed in Appendix E.1 E.2 and E.3. Additionally, we provide a hyperparameter analysis in Appendix F and an error analysis in Appendix G. The additional cost and efficiency of the methods are discussed in Appendix D.

## 3.2 OVERALL RESULTS

We present the results for Question Answering and Retrieval in Table 1 and Table 2, respectively. We also compare performance of single-granularity and multi-granularity in Appendix C. Below, we provide analysis of these results.

**Question Answering Results.** As presented in Table 1, MemGAS consistently outperforms other methods across most datasets and evaluation metrics. Unlike Full History, which introduces noise by utilizing all historical context, MemGAS excels by effectively consolidating and retrieving only the most relevant memories. Other baselines, such as RecurSum and SeCom, although operating at specific granularities, lack the capability to integrate multi-granular associations between memory, resulting in suboptimal outcomes. Besides, methods like HippoRAG 2 and RAPTOR, while establishing connections between memory units, fail to construct multi-granular relationships and selection mechanisms effectively, limiting their performance. MemGAS performs better through its dynamic construction and adaptive router of multi-granular memory units and redundancy filtering, emphasizing the critical role of association and selection in memory management. In terms of efficiency, MemGAS maintains competitive token usage and achieves stronger QA quality while retaining comparable—or even lower—latency than several baselines such as A-Mem and RecurSum, making it both more accurate and more efficient overall.

**Retrieval Results.** As shown in Table 2, our MemGAS demonstrates outstanding performance across all datasets, consistently achieving the highest Recall and NDCG metrics. These results highlight the effectiveness and robustness of our approach, addressing key challenges in long-term memory construction and retrieval, and ensuring that queries are matched with the most relevant context.

Table 1: QA performance. Contriever is used as the retrieval backbone for all baselines (except for Full History and MPNet), with GPT4o-mini serving as the generator. The **bold** values indicate the best performance, while underlined values mark the second-best across each metric. All evaluation metrics follow a 'higher is better' convention. '**4o-J**' denotes GPT4o-as-Judge, '**B-4**' represents BLEU4, '**R-n**' refers to ROUGE-n, and '**BS**' denotes BERTScore. Avg. Tokens and Avg. Latency both reports the average computational cost per query; Avg. Tokens measure LLM API token consumption, while Avg. Latency reflects the end-to-end response time. Latency is measured in seconds.

| Model | 4o-J | F1 | B-4 | R-1 | R-2 | R-L | BS | Avg. Tokens | Avg. Latency |
|---|---|---|---|---|---|---|---|---|---|
| *LongMemEval-s* | | | | | | | | | |
| Full History | 50.60 | 11.48 | 1.40 | 12.10 | 5.47 | 10.85 | 83.07 | 103,137 | 9.39 |
| MPNet (Song et al., 2020) | 53.20 | 13.96 | 2.21 | 14.49 | 6.78 | 12.93 | 83.72 | 8,173 | 1.82 |
| Contriever (Izacard et al., 2021) | 55.40 | 13.78 | 2.21 | 14.46 | 6.93 | 12.89 | 83.70 | 8,286 | 1.85 |
| MPC (Lee et al., 2023) | 53.80 | 13.60 | 1.74 | 14.27 | 6.49 | 12.95 | 83.49 | 8,457 | 2.66 |
| RecurSum (Wang et al., 2025) | 35.40 | 12.29 | 2.09 | 13.01 | 5.55 | 11.52 | 83.60 | 8,853 | 1.94 |
| SeCom (Pan et al., 2025) | 56.00 | 12.95 | 2.25 | 13.80 | 6.09 | 11.93 | 83.51 | 2,741 | 1.67 |
| HippoRAG 2(Gutiérrez et al., 2025) | 57.60 | 14.73 | 2.15 | 15.30 | 7.36 | 13.83 | 83.86 | 8,530 | 4.51 |
| RAPTOR (Sarthi et al., 2024) | 32.20 | 12.08 | 1.90 | 12.73 | 5.82 | 11.25 | 83.50 | 6,254 | 2.25 |
| A-Mem (Xu et al., 2025b) | 55.60 | 13.73 | 2.11 | 14.82 | 6.81 | 12.98 | 83.88 | 9,018 | 2.59 |
| **MemGAS (Ours)** | **60.20** | **20.38** | **4.22** | **21.05** | **10.47** | **19.47** | **85.21** | 8,829 | 2.55 |
| *LongMemEval-m* | | | | | | | | | |
| Full History | 12.20 | 5.70 | 0.78 | 6.27 | 2.08 | 5.28 | 81.62 | 128,000 | 12.88 |
| MPNet (Song et al., 2020) | 37.80 | 10.76 | 1.70 | 11.46 | 4.76 | 10.03 | 83.09 | 8,352 | 1.83 |
| Contriever (Izacard et al., 2021) | 42.80 | 11.88 | 1.66 | 12.56 | 5.63 | 11.02 | 83.28 | 8,467 | 1.92 |
| MPC (Lee et al., 2023) | 37.80 | 11.28 | 1.37 | 11.93 | 5.12 | 10.57 | 82.98 | 8,428 | 2.75 |
| RecurSum (Wang et al., 2025) | 23.80 | 10.04 | 1.70 | 10.89 | 4.26 | 9.21 | 83.12 | 8,927 | 1.99 |
| SeCom (Pan et al., 2025) | 42.80 | 11.33 | 1.79 | 12.03 | 5.07 | 10.49 | 83.36 | 2,821 | 1.63 |
| **MemGAS (Ours)** | **45.40** | **16.85** | **3.39** | **17.60** | **8.25** | **16.14** | **84.69** | 8,852 | 2.45 |
| *LoCoMo* | | | | | | | | | |
| Full History | 33.43 | 12.23 | 1.84 | 12.70 | 5.66 | 11.73 | 84.07 | 20,078 | 4.92 |
| MPNet (Song et al., 2020) | 38.07 | 14.44 | 2.35 | 14.90 | 6.83 | 13.90 | 84.42 | 2,472 | 1.29 |
| Contriever (Izacard et al., 2021) | 40.33 | 15.66 | 2.67 | 16.01 | 7.68 | 15.00 | 84.65 | 2,348 | 1.24 |
| MPC (Lee et al., 2023) | 40.38 | 14.81 | 1.99 | 15.10 | 6.83 | 14.13 | 84.43 | 2,683 | 1.95 |
| RecurSum (Wang et al., 2025) | 22.56 | 9.14 | 0.99 | 9.82 | 3.38 | 8.98 | 83.45 | 3,074 | 1.58 |
| SeCom (Pan et al., 2025) | 44.21 | 13.79 | 2.30 | 14.28 | 6.17 | 13.30 | 84.04 | 1,021 | 1.02 |
| HippoRAG 2(Gutiérrez et al., 2025) | **45.62** | 16.66 | 2.91 | 17.01 | 8.27 | 15.93 | 84.88 | 2,991 | 3.56 |
| RAPTOR (Sarthi et al., 2024) | 31.72 | 14.55 | 2.88 | 15.09 | 7.49 | 14.18 | 84.48 | 1,931 | 1.73 |
| A-Mem (Xu et al., 2025b) | 40.81 | 14.72 | 2.83 | 16.22 | 7.71 | 14.89 | 84.72 | 3,042 | 1.98 |
| **MemGAS (Ours)** | 41.07 | **17.66** | **3.61** | **18.00** | **8.93** | **16.99** | **85.13** | 2,825 | 1.88 |
| *LongMTBench+* | | | | | | | | | |
| Full History | 67.44 | 36.07 | 11.32 | 37.90 | 20.51 | 29.25 | 87.81 | 19,194 | 4.72 |
| MPNet (Song et al., 2020) | 63.89 | 36.09 | 11.26 | 38.39 | 20.58 | 28.86 | 87.84 | 12,143 | 3.21 |
| Contriever (Izacard et al., 2021) | 63.54 | 36.30 | 11.59 | 38.17 | 21.65 | 29.67 | 87.90 | 11,941 | 3.27 |
| MPC (Lee et al., 2023) | 61.81 | 31.52 | 7.97 | 33.56 | 17.27 | 25.20 | 86.51 | 12,289 | 3.98 |
| RecurSum (Wang et al., 2025) | 24.65 | 26.58 | 6.91 | 29.23 | 11.93 | 20.90 | 86.11 | 13,527 | 3.41 |
| SeCom (Pan et al., 2025) | 64.58 | 36.68 | 12.01 | 38.81 | 21.44 | 29.65 | 87.88 | 4,714 | 2.68 |
| HippoRAG 2(Gutiérrez et al., 2025) | 63.54 | 35.64 | 11.05 | 37.61 | 20.37 | 28.76 | 87.70 | 13,583 | 6.14 |
| RAPTOR (Sarthi et al., 2024) | 59.72 | 37.69 | 13.47 | 40.08 | 21.68 | 30.88 | 88.38 | 10,631 | 3.47 |
| A-Mem (Xu et al., 2025b) | 65.73 | 36.82 | 11.36 | 38.92 | 20.88 | 29.14 | 87.92 | 13,735 | 3.95 |
| **MemGAS (Ours)** | **69.44** | **41.49** | **15.62** | **43.69** | **24.45** | **34.66** | **88.96** | 12,873 | 3.85 |

## 3.3 ABLATION STUDY

The ablation study in Table 3 demonstrates the significance of each component in enhancing both QA and retrieval performance. Individually removing GMM, PPR, MA, or the Router results in

Table 2: Retrieval Performance. All methods are based on Contriever as the retriever, except for MPNet.

| Model | Recall@3 | NDCG@3 | Recall@5 | NDCG@5 | Recall@10 | NDCG@10 |
|---|---|---|---|---|---|---|
| *LongMemEval-s* | | | | | | |
| MPNet (Song et al., 2020) | 66.17 | 75.47 | 76.38 | 78.29 | 85.11 | 80.63 |
| Contriever (Izacard et al., 2021) | 71.06 | 79.72 | 81.28 | 82.47 | 90.00 | 84.29 |
| RecurSum (Wang et al., 2025) | 67.23 | 78.33 | 79.79 | 81.76 | 87.66 | 83.28 |
| MPC (Lee et al., 2023) | 60.00 | 70.90 | 68.09 | 73.27 | 80.00 | 76.59 |
| SeCom (Pan et al., 2025) | 71.06 | 80.88 | 80.43 | 83.08 | 89.15 | 85.11 |
| HippoRAG 2(Gutiérrez et al., 2025) | 75.53 | 85.44 | 84.68 | 87.32 | 91.28 | 88.73 |
| **MemGAS** (Ours) | **78.51** | **86.83** | **88.94** | **88.77** | **94.47** | **89.96** |
| *LongMemEval-m* | | | | | | |
| MPNet (Song et al., 2020) | 37.02 | 48.68 | 45.74 | 52.51 | 61.28 | 56.55 |
| Contriever (Izacard et al., 2021) | 44.26 | 56.11 | 53.40 | 59.44 | 65.96 | 62.74 |
| RecurSum (Wang et al., 2025) | 23.19 | 32.10 | 31.70 | 36.99 | 43.62 | 41.40 |
| MPC (Lee et al., 2023) | 35.96 | 47.51 | 42.55 | 50.36 | 54.26 | 53.88 |
| SeCom (Pan et al., 2025) | 44.26 | 56.61 | 55.32 | 60.76 | 66.60 | 63.78 |
| **MemGAS** (Ours) | **51.06** | **61.36** | **63.62** | **66.07** | **77.02** | **69.46** |
| *LoCoMo* | | | | | | |
| MPNet (Song et al., 2020) | 45.92 | 47.71 | 53.98 | 51.79 | 68.58 | 56.88 |
| Contriever (Izacard et al., 2021) | 49.90 | 52.15 | 58.26 | 56.29 | 71.80 | 60.92 |
| RecurSum (Wang et al., 2025) | 47.23 | 48.99 | 59.01 | 54.58 | 74.97 | 60.07 |
| MPC (Lee et al., 2023) | 49.50 | 51.47 | 57.45 | 55.53 | 71.85 | 60.47 |
| SeCom (Pan et al., 2025) | 55.24 | 57.90 | 64.80 | 62.36 | 78.30 | 66.97 |
| HippoRAG 2(Gutiérrez et al., 2025) | 56.60 | 58.37 | 65.06 | 62.50 | 78.05 | 66.79 |
| **MemGAS** (Ours) | **57.30** | **58.76** | **67.32** | **63.62** | **81.82** | **68.42** |

Table 3: Ablation Study on LongMemeval-s for Gaussian Mixture Model (GMM), Personalized PageRank (PPR), Granularity Router and Memory Association (MA). The w/o MA setting is equivalent to w/o GMM and PPR. QA performance is evaluated using top 3 retrieved results. R@n means Recall@n. The (Δ) represents the latency introduced by each module.

| Method | QA Performance | | | | | Retrieval Performance | | | |
|---|---|---|---|---|---|---|---|---|---|
| | GPT4o-J | F1 | RogueL | Avg. Tokens | Total Latency (s) | R@3 | R@5 | R@10 | Retrieval Latency (s) |
| **MemGAS** | **60.20** | **20.38** | **19.47** | 8,829 | 2.5534 | **78.51** | **88.94** | **94.47** | 0.0239 |
| w/o GMM | 57.20 | 19.49 | 18.68 | 8,820 | 2.5506(Δ0.0028) | 76.38 | 85.53 | 91.28 | 0.0232(Δ0.0007) |
| w/o PPR | 56.60 | 19.76 | 18.85 | 8,772 | 2.5449(Δ0.0085) | 75.96 | 85.96 | 90.64 | 0.0194(Δ0.0045) |
| w/o MA | 56.80 | 17.69 | 19.00 | 8,734 | 2.5418(Δ0.0116) | 74.89 | 85.74 | 91.49 | 0.0182(Δ0.0057) |
| w/o Router | 56.60 | 18.88 | 18.62 | 8,763 | 2.5471(Δ0.0063) | 75.53 | 85.74 | 92.34 | 0.0216(Δ0.0023) |
| w/o All | 55.40 | 13.78 | 12.89 | 8,701 | 2.5343(Δ0.0191) | 71.06 | 81.28 | 90.00 | 0.0160(Δ0.0079) |

consistent performance degradation, validating their essential contributions. Notably, the combined absence of all components leads to the most significant drop, with the F1 score plummeting from 20.38 to 13.78, and Recall@3 decreasing from 78.51 to 71.06. This highlights the importance of all modules in improving overall performance. Moreover, **the latency introduced by these modules is minimal**, with the highest latency increase being only 0.0191 seconds for QA and 0.0079 seconds for retrieval. We also found that LLM API calls account for over 98% of the overall end-to-end latency, indicating that they are by far the primary contributor to the system's response time, and the overhead from our modules is acceptable in practice. This demonstrates that the proposed architecture achieves a remarkable balance between enhanced performance and computational efficiency.

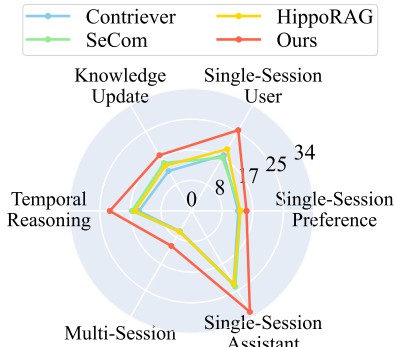

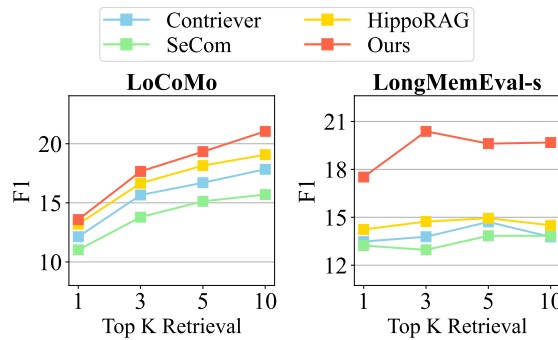

Figure 3: Comparison of F1 Across Different Query Types in LongMemEval-s.

Figure 4: Comparison on Different Top-K Retrieval.

## 3.4 DETAILED COMPARISON ANALYSIS

In this section, we present a comprehensive analysis comparing our approach with baselines across different query types and different Top-K retrieval settings.

**Comparison on different query types.** The results in Figure 5 highlight performance across different query types. Our MemGAS consistently demonstrates superior performance, particularly excelling in *multi-hop retrieval* on LoCoMo and *multi-session* on LongMemEval-s. This suggests that memory association mechanism in MemGAS effectively identifies highly relevant sessions, enabling enhanced multi-hop reasoning and better integration of knowledge. Additionally, our method performs well across single-session, temporal reasoning, and knowledge update, demonstrating its strength in addressing both simple and complex query types. We also provide a case study for different query types in Appendix K.1, and comparison with other datasets and metrics in Appendix E.3.

**Comparison on different Top-K retrieval.** In downstream QA tasks, Top-K retrieved memories are directly fed as context to LLM generator, where increasing $K$ improves coverage but may introduce noise when $K$ becomes too large. The results shown in Figure 4 demonstrate the performance of various Top-K retrieval settings. On the LoCoMo and LongMemEval-s datasets, our model consistently surpasses baseline methods in F1 scores as Top-K increases from 1 to 10. This trend highlights that retrieving more relevant context significantly enhances accuracy. Note that on the LongMemEval-s dataset, while F1 scores initially improve at lower Top-K values, performance declines at higher Top-K levels, suggesting that the longer context introduces noise, which negatively impacts the model's effectiveness.

**Comparison w/ and w/o Filter Setting.** To ensure fairness when comparing MemGAS with baselines that may not employ LLM-based filtering, we provide a detailed breakdown of token usage for *online per-query cost*. Table 4 reports token consumption with and without filtering, as well as the corresponding QA performance on LongMemEval-s using GPT4o-J. Overall, MemGAS exhibits competitive or lower online token usage compared to representative structured baselines such as HippoRAG2, while consistently achieving higher QA accuracy. The filtering module introduces only a small additional overhead (approximately 200–300 tokens per query), yet it improves answer precision and benefits downstream reasoning quality. Moreover, as shown in Appendix D.1, MemGAS requires noticeably less LLM involvement during memory construction compared to multi-stage structured approaches (e.g., HippoRAG2), further enhancing its overall efficiency.

## 4 RELATED WORK

**Retrieval Methods.** Recent advancements in retrieval methods have greatly enhanced information retrieval performance(Robertson et al., 2009; Song et al., 2020; Santhanam et al., 2021; Zhang et al.). BM25 (Robertson et al., 2009), a classic sparse retrieval method, uses a probabilistic model to improve query relevance. In contrast, dense retrieval methods excel at capturing semantic similarity. For instance, DPR (Karpukhin et al., 2020) employs dual-encoder models to encode queries and documents into dense vectors for efficient similarity search. Similarly, Contriever (Izacard et al.,

Table 4: Comparison of LLM token usage and QA performance between MemGAS and representative baselines, under both filtering and non-filtering settings. The reported numbers include (i) average online per-query token cost and (ii) GPT4o-J QA score on LongMemEval-s. Avg. Tokens per query denotes online QA-stage token consumption: tokens for LLM filtering + tokens for QA generation.

| Category | Setting | MemGAS | HippoRAG2 | RAPTOR | SeCom | RecurSum |
|----------|---------|--------|-----------|--------|-------|----------|
| Avg. Tokens per query | With filter | 8,829 | 8,911 | 6,617 | 3,015 | 9,176 |
| Avg. Tokens per query | Without filter | 8,481 | 8,530 | 6,254 | 2,741 | 8,853 |
| QA performance | With filter | **60.2** | 58.4 | 33.2 | 56.6 | 36.2 |
| QA performance | Without filter | **59.4** | 57.6 | 32.2 | 56.0 | 35.4 |

2021) leverages contrastive learning to enhance semantic understanding. Advanced methods like E5 (Wang et al., 2022), BGE (Luo et al., 2024), and GTE (Li et al., 2023) further utilize pre-trained or fine-tuned transformers for robust and efficient semantic retrieval.

**Long Term Memory Management.** With the development of LLMs, the user-assistant conversation becomes longer and contains various topics (Chhikara et al., 2025; Li et al., 2025c; Wang & Chen, 2025; Lu et al., 2023; Du et al., 2024; Qian et al., 2025; Packer et al., 2023; Zhang et al., 2025a; 2024a; Wu et al., 2025b; Du et al., 2024; Kirmayr et al., 2025), which introduces challenges for preserving the user's long-term memory. Management in long-term memory often involves segmentation (Pan et al., 2025), summary (Kim et al., 2024), compression (Chen et al., 2025b), forgetting and updating (Bae et al., 2022; Wang et al., 2025; Zhong et al., 2024). For example, several approaches (Wang et al., 2025; Sarthi et al., 2024; Team, 2023; Liu et al., 2023a; Lu et al., 2023) focus on generating memory summaries as records to enable more accurate retrieval. Besides, some methods (Pan et al., 2025; Lee et al., 2024; Xu et al., 2023; Chen et al., 2025a) leverage compression techniques to reduce memory size while preserving essential information. The forgetting mechanisms (Zhong et al., 2024; Jia et al., 2024) address the need to remove obsolete or irrelevant memories while maintaining model performance. Some integrated methods (Tan et al., 2025; Ong et al., 2025; Li et al., 2024a) combine memory retention, update to enable personalized and contextually relevant long-term dialogue. However, existing approaches typically focus on single-granularity segmentation strategies, such as session/turn, to organize and manage long-term memory. Whereas our work leverages multi-granular information for better adaptive memory selection.

**Structural Memory Management.** Additionally, existing works employ structured paradigms for memory or knowledge base organization (Xu et al., 2024; Chen et al., 2024; Li et al., 2024c; Zhang et al., 2025a; 2024a; Wu et al., 2025b; He et al., 2024a; Zhang et al., 2024b; Xu et al., 2022; Rasmussen et al., 2025). HippoRAG (Gutiérrez et al.; 2025) builds entity-centric knowledge graphs inspired by hippocampal indexing theory (Teyler & DiScenna, 1986), while Graph-CoT (Jin et al., 2024) and G-Retriever (He et al., 2024b) integrate graph reasoning for interactive retrieval or generation, whereas LightRAG (Guo et al., 2024) and GraphRAG (Edge et al., 2024) optimize retrieval and summary via graph structures. MemTree and RAPTOR (Rezazadeh et al., 2024; Sarthi et al., 2024) utilize recursive embedding, clustering, and summarization of text chunks to construct a hierarchical tree, while StructRAG (Li et al., 2024c) enhances reasoning by leveraging multiple structured formats. While existing methods focus on single-granularity memory modeling, they lack cross-granularity interactions. In contrast, our MemGAS utilizes multi-granularity association and adaptive selection to construct consolidated memory structures and optimize retrieval efficiency.

## 5 CONCLUSION

In this paper, we proposed MemGAS, a novel framework for long-term memory construction and retrieval that integrates multi-granular memory units and enables adaptive selection and retrieval. By leveraging human-inspired memory mechanisms through Gaussian Mixture Models and an entropy-based multi-granularity router, MemGAS effectively addresses challenges of memory connection and selection. Experimental results across four benchmarks demonstrate that MemGAS significantly outperforms state-of-the-art baselines in both QA and retrieval tasks, highlighting its robustness and superiority in managing long-term memory for conversational agents.

## ETHICS STATEMENT

This work adheres to ethical research practices by ensuring that all experiments and datasets used are publicly available and utilized in accordance with their licenses. The proposed MemGAS framework is designed to enhance conversational agents responsibly, with a focus on improving user experience while minimizing the risk of harm, such as generating misinformation. No sensitive or proprietary data was used during the research process, and the methodology prioritizes transparency and accountability in memory retrieval and usage.

## REPRODUCIBILITY STATEMENT

We are committed to ensuring the reproducibility of our work. To this end, we have provided the code in an anonymous repository. The repository includes a well-documented README file with instructions to replicate our experiments. Additionally, we have detailed implementation specifics in this manuscript, and hyperparameter tuning is comprehensively described in Appendix F. Furthermore, we have included the complete set of prompts required for the experiments, ensuring that all components of our methodology can be accurately reproduced. We encourage the community to leverage these resources to build upon our work.

## ACKNOWLEDGMENTS

This work was supported in part by the grants from National Science and Technology Major Project (No. 2023ZD0121104), and National Natural Science Foundation of China (No.62222213, U22B2059, No.62502404). This research was also partially supported by Hong Kong Research Grants Council (Research Impact Fund No.R1015-23, Collaborative Research Fund No.C1043-24GF, General Research Fund No.11218325), Institute of Digital Medicine of City University of Hong Kong (No.9229503), Huawei (Huawei Innovation Research Program).

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

# Technical Appendix

## TABLE OF CONTENTS

## A    DATASETS STATISTICS

Table 5: Dataset statistics. The term 'Avg.' (e.g., Avg. Sessions) represents the average number corresponding to each conversation.

| Dataset | LoCoMo | Long-MT-Bench+ | LongMemEval-s | LongMemEval-m |
|---|---|---|---|---|
| Total Conversations | 10 | 11 | 500 | 500 |
| Avg. Sessions | 27.2 | 4.9 | 50.2 | 501.9 |
| Avg. Query | 198.6 | 26.2 | 1.0 | 1.0 |
| Avg. Token | 20,078.9 | 19,194.8 | 103,137.4 | 1,019,116.7 |
| Sessions Dates | ✓ | ✗ | ✓ | ✓ |
| Retrieval Ground-Truth | ✓ | ✗ | ✓ | ✓ |
| QA Ground-Truth | ✓ | ✓ | ✓ | ✓ |
| Conversation Subject | User-User | User-AI | User-AI | User-AI |

We evaluate the proposed method with the following benchmarks:

- **LongMemEval-s** (Wu et al., 2025a) and **LongMemEval-m** (Wu et al., 2025a) datasets are benchmarks designed to evaluate long-term memory in chat assistants. LongMemEval-s involves 50 sessions per question with an average of 115k tokens, making it a compact yet challenging dataset. LongMemEval-m, on the other hand, spans 500 sessions per question, resulting in avg. 1.5 million tokens per conversation, offering a more extensive evaluation setting. Both datasets require AI systems to handle user-AI dialogues, dynamic memorization, and historical consistency. The dataset includes several query types: *(1) Knowledge Update*, testing whether the model can track and reason about changes in the user's personal information over time; *(2) Temporal Reasoning*, focusing on explicit dates and inferred time references; *(3) Single-Session User*, which checks recall of user details within one session; *(4) Single-Session Preference*, evaluating whether the model can use preferences shared in a session; *(5) Single-Session Assistant*, testing recall of assistant-provided information; *(6) Multi-Session*, which assesses reasoning that spans multiple sessions.

- **LoCoMo** (Maharana et al., 2024) dataset evaluates long-term memory in AI through lengthy conversations, averaging 300 turns, 9,000 tokens, and up to 35 sessions. LoCoMo, the publicly available subset of the original paper, includes 10 high-quality, long conversations with 27.2 sessions and 20,000 tokens on average. The goal of LoCoMo is to evaluate long-context LLMs and RAG systems, which, while improving memory performance, still fall short of human capabilities—particularly in temporal reasoning—underscoring the challenges of understanding long-range dependencies. The dataset includes several query types: *(1) Open Domain Knowledge*, requiring integrating speaker-provided facts with external or commonsense knowledge; *(2) Temporal Reasoning*, involving chronological inference; *(3) Single-Hop Retrieval*, solvable via a single session; *(4) Adversarial*, which intentionally misleads and requires the model to avoid incorrect conclusions; *(5) Multi-Hop Retrieval*, requiring synthesis across multiple sessions.

- **Long-MT-Bench+** (Pan et al., 2025) dataset is reconstructed from MT-Bench+ (Lu et al., 2023) by incorporating long-range questions to address the limited QA pairs and short dialogues. It merges five consecutive sessions into a single long-term conversation, improving its suitability for evaluating memory mechanisms. The dataset features 11 conversations with an average of 4.9 sessions and 19194.8 tokens. Unlike LoCoMo, it focuses on user-AI interactions, excluding session dates and retrieval ground truth.

## B    COMPARISON OF METHODS STRUCTURE WITH BASELINES

We compare the model structures of each baseline **(provided in parentheses)** and offer concise summaries to highlight the differences in their memory construction and retrieval mechanisms.

- **MPNet (Vector Base)**: Uses permuted language modeling with auxiliary positional information to model token dependencies while seeing full-sentence position signals.
- **Contriever (Vector Base)**: Trains dense retrievers via unsupervised contrastive learning for generalizable text embeddings, including cross-lingual retrieval.

- **MPC (Memory Pool)**: Composes LLM modules with few-shot prompting, chain-of-thought, and external memory to maintain long-term conversational consistency without fine-tuning.

- **RecurSum (Recursive Summary)**: Recursively generates and updates dialogue summaries as memory, using prior memory and new context to maintain coherence.

- **SeCom (Semantic Segment)**: Segments conversations into topically coherent units and applies compression-based denoising to build and retrieve from segment-level memory.

- **RAPTOR (Hierarchical Tree)**: Recursively embeds, clusters, and summarizes text to form a hierarchical summary tree, enabling retrieval at multiple abstraction levels.

- **HippoRAG2 (Knowledge Graph)**: Augments RAG with knowledge-graph traversal, integrating passages and online LLM reasoning for associative retrieval.

- **A-Mem (Memory Note)**: Implements a Zettelkasten-inspired memory: creates structured notes with attributes, indexes and links them, and updates existing notes as new ones arrive.

- **MemGAS (Ours; Multi-Granularity Memory)**: Builds multi-granularity memory units, associates them via Gaussian Mixture Models, and uses an entropy-based router plus LLM filtering for adaptive retrieval.

## C  COMPREHENSIVE COMPARISON OF SINGLE-GRANULARITY AND MULTI-GRANULARITY

### C.1  QA PERFORMANCE.

The results in the Table 6 demonstrate the significant advantages of multi-granularity methods over single-granularity approaches. Across all datasets, multi-granularity methods outperform single-granularity methods in key evaluation metrics such as F1, BLEU4, ROUGE, and BERTScore. For example, in the LongMemEval-m dataset, the best-performing single-granularity method (Turn-level) achieves an F1 score of 12.26, whereas the multi-granularity method achieves significantly higher F1 scores of 12.51 and 16.85. Similarly, in the LongMTBench+ dataset, the multi-granularity method achieves F1 scores of 37.16 and 41.49, outperforming the best single-granularity method, which scores 36.67. These results highlight the effectiveness of multi-granularity approaches in integrating information from different granularities, providing a more comprehensive understanding and significantly improving QA task performance.

Moreover, the proposed method, MemGAS reveals its superiority over simple multi-granularity combination methods. Across all datasets, MemGAS consistently achieves better performance not only in F1 scores but also across other evaluation metrics. For instance, in the LongMemEval-s dataset, MemGAS achieves an F1 score of 20.38, surpassing the combination method's score of 14.59. Similarly, in the LongMTBench+ dataset, MemGAS achieves an F1 score of 41.49, compared to 37.16 for the combination method. These results demonstrate that MemGAS leverages a more sophisticated mechanism to capture the relationships and complementarities among different granularities. This enables MemGAS to achieve greater robustness and generalization, making it particularly effective for complex QA tasks.

### C.2  RETRIEVAL PERFORMANCE

The results in Table 7 clearly demonstrate the advantages of the multi-granularity approach in retrieval tasks compared to single-granularity methods. Across all datasets, multi-granularity methods consistently achieve higher scores in key retrieval metrics such as Recall and NDCG. For example, in the LongMemEval-s dataset, the best single-granularity method (Turn-level) achieves Recall@10 of 91.91 and NDCG@10 of 86.73. However, the simplest multi-granularity combination approach further improves these scores to 91.91 and 88.03, respectively, demonstrating its superior retrieval effectiveness. Similarly, in the LongMemEval-m dataset, the Recall@10 and NDCG@10 of the multi-granularity approach reached at least 73.83 and 68.66, respectively, which significantly exceeded the best results of the single-granularity approach (Recall@10 = 70.21 and NDCG@10 = 63.43, Turn-level). These findings underline the strength of multi-granularity approaches in leveraging diverse levels of information to retrieve more relevant results and improve overall retrieval quality.

Table 6: QA performance of Single-Granularity and Multi-Granularity.

| Granularity | GPT4o-J | F1 | BLEU4 | Rouge1 | Rouge2 | RougeL | BERTScore |
|---|---|---|---|---|---|---|---|
| **_LongMemEval-s_** | | | | | | | |
| _Single-Granularity_ | | | | | | | |
| Session-level | 55.40 | 13.78 | 2.21 | 14.46 | 6.93 | 12.89 | 83.70 |
| Turn-level | _57.60_ | _14.94_ | _2.50_ | _15.53_ | _7.50_ | _14.12_ | 83.93 |
| Summary-level | 28.80 | 10.66 | 1.78 | 11.51 | 4.61 | 9.96 | 83.26 |
| Keyword-level | 17.20 | 9.05 | 1.61 | 9.82 | 3.61 | 8.20 | 82.84 |
| _Multi-Granularity_ | | | | | | | |
| Combination | 56.60 | 14.59 | 2.40 | 15.19 | 7.22 | 13.70 | _83.96_ |
| **MemGAS** | **60.20** | **20.38** | **4.22** | **21.05** | **10.47** | **19.47** | **85.21** |
| **_LongMemEval-m_** | | | | | | | |
| _Single-Granularity_ | | | | | | | |
| Session-level | 42.80 | 11.88 | 1.66 | 12.56 | 5.63 | 11.02 | 83.28 |
| Turn-level | 42.60 | 12.26 | 1.79 | 12.93 | 5.67 | 11.45 | 83.43 |
| Summary-level | 24.40 | 10.11 | 1.68 | 11.02 | 4.16 | 9.41 | 83.17 |
| Keyword-level | 15.00 | 8.29 | 1.54 | 9.16 | 3.26 | 7.43 | 82.72 |
| _Multi-Granularity_ | | | | | | | |
| Combination | **46.40** | _12.51_ | _1.96_ | _13.08_ | _6.13_ | _11.70_ | _83.49_ |
| **MemGAS** | _45.40_ | **16.85** | **3.39** | **17.60** | **8.25** | **16.14** | **84.69** |
| **_LoCoMo_** | | | | | | | |
| _Single-Granularity_ | | | | | | | |
| Session-level | 40.33 | 15.66 | 2.67 | 16.01 | 7.68 | 15.00 | 84.65 |
| Turn-level | _42.09_ | _16.10_ | _2.80_ | _16.42_ | _8.13_ | _15.36_ | _84.70_ |
| Summary-level | 19.08 | 9.78 | 0.83 | 10.38 | 3.28 | 9.41 | 83.69 |
| Keyword-level | 14.85 | 7.74 | 0.64 | 8.40 | 2.20 | 7.62 | 83.28 |
| _Multi-Granularity_ | | | | | | | |
| Combination | **42.80** | 15.93 | 2.60 | 16.27 | 7.76 | 15.26 | 84.69 |
| **MemGAS** | 40.08 | **17.66** | **3.61** | **18.00** | **8.93** | **16.99** | **85.13** |
| **_LongMTBench+_** | | | | | | | |
| _Single-Granularity_ | | | | | | | |
| Session-level | 63.54 | 36.30 | 11.59 | 38.17 | 21.65 | 29.67 | 87.90 |
| Turn-level | 67.01 | 36.67 | _11.91_ | 39.06 | 20.88 | 30.13 | _88.04_ |
| Summary-level | 21.18 | 28.30 | 8.37 | 31.12 | 13.11 | 22.55 | 86.74 |
| Keyword-level | 20.83 | 28.05 | 8.90 | 30.92 | 12.80 | 22.39 | 86.78 |
| _Multi-Granularity_ | | | | | | | |
| Combination | _67.01_ | _37.16_ | 11.58 | _39.11_ | _21.95_ | _30.24_ | 88.00 |
| **MemGAS** | **69.44** | **41.49** | **15.62** | **43.69** | **24.45** | **34.66** | **88.96** |

"Combination" denotes that texts from all granularities are directly concatenated into a single string, and encoded to one embedding for retrieval. This naive merging lacks adaptive weighting or structure. The proposed method, MemGAS further demonstrates its superiority over simple multi-granularity combination methods. In all datasets, MemGAS consistently achieves better results. For example, in the LongMemEval-s dataset, MemGAS achieves Recall@10 of 94.47 and NDCG@10 of 89.96, surpassing the combination method's scores of 91.91 and 88.03. Similarly, in the LoCoMo dataset, MemGAS achieves Recall@10 of 81.07 and NDCG@10 of 68.24, outperforming the combination method's scores of 78.70 and 65.64. These improvements demonstrate that MemGAS is not merely aggregating information from multiple granularities but instead employs a more advanced mechanism to capture the interdependencies and complementarities among different granularities. This enables MemGAS to deliver robust and versatile performance, making it highly effective for complex retrieval tasks that require the integration of multi-granularity information.

## C.3 HOW MULTI-GRANULARITY ROUTER WORKS?

In this section, we analyze how the proposed Multi-granularity Router works. "Optimal Selection" is an oracle-style upper bound: for each query, it picks the single best-performing granularity among

Table 7: Retrieval Performance of Single-Granularity and Multi-Granularity.

| Granularity | Recall@3 | NDCG@3 | Recall@5 | NDCG@5 | Recall@10 | NDCG@10 |
|---|---|---|---|---|---|---|
| *LongMemEval-s* | | | | | | |
| *Single-Granularity* | | | | | | |
| Session-level | 71.06 | 79.72 | 81.28 | 82.47 | 90.00 | 84.29 |
| Turn-level | 73.62 | 82.47 | 84.68 | 84.94 | 91.91 | 86.73 |
| Summary-level | 70.43 | 80.53 | 80.00 | 82.38 | 88.30 | 84.28 |
| Keyword-level | 62.98 | 74.69 | 74.68 | 77.76 | 82.34 | 79.50 |
| *Multi-Granularity* | | | | | | |
| Combination | 76.17 | 84.68 | 87.23 | 87.00 | 91.91 | 88.03 |
| **MemGAS** | **78.51** | **86.83** | **88.94** | **88.77** | **94.47** | **89.96** |
| *LongMemEval-m* | | | | | | |
| *Single-Granularity* | | | | | | |
| Session-level | 44.26 | 56.11 | 53.40 | 59.44 | 65.96 | 62.74 |
| Turn-level | 44.68 | 55.06 | 57.66 | 60.03 | 70.21 | 63.43 |
| Summary-level | 41.06 | 54.68 | 52.34 | 58.64 | 65.53 | 62.31 |
| Keyword-level | 35.11 | 48.22 | 43.62 | 52.02 | 57.87 | 55.75 |
| *Multi-Granularity* | | | | | | |
| Combination | 50.85 | **61.50** | 60.00 | 64.82 | 73.83 | 68.66 |
| **MemGAS** | **51.06** | 61.36 | **63.62** | **66.07** | **77.02** | **69.46** |
| *LoCoMo* | | | | | | |
| *Single-Granularity* | | | | | | |
| Session-level | 49.90 | 52.15 | 58.26 | 56.29 | 71.80 | 60.92 |
| Turn-level | 52.77 | 54.09 | 64.30 | 59.64 | 80.31 | 65.07 |
| Summary-level | 47.89 | 48.96 | 58.21 | 53.94 | 74.12 | 59.61 |
| Keyword-level | 29.05 | 29.72 | 40.33 | 35.46 | 65.11 | 44.11 |
| *Multi-Granularity* | | | | | | |
| Combination | 53.98 | 55.89 | 63.80 | 60.61 | 78.70 | 65.64 |
| **MemGAS** | **57.45** | **58.84** | **67.12** | **63.60** | **81.07** | **68.24** |

the four. Conceptually, it reflects the upper performance achievable if one could choose the ideal granularity per query. **Our findings reveal that different levels of granularity exhibit distinct preferences on the query type, and our router effectively adapts by selecting the suitable granularity strategy**. Table 8 presents the retrieval performance (Recall@3) across various query types and granularities on the LongMemEval-m dataset. The results show that different query types favor different granularities: for instance, temporal-reasoning queries benefit most from session-level granularity, knowledge-update queries achieve better performance with turn-level granularity, and single-session-preference queries perform best with summary-level granularity. Notably, our Granularity Router (MemGAS) adaptively integrates multiple granularities, achieving retrieval performance that closely approaches the upper bound defined by Optimal Selection. This highlights the router's ability to dynamically identify and apply the most effective granularity for each query type, bridging the gap between fixed granularity approaches and optimal strategies.

## C.4 How PPR affects the retrieval results?

To further understand how PPR reshapes the retrieval behavior, we conduct a detailed analysis comparing the top-k results before and after applying PPR. As shown in Table 9, PPR leads to substantial re-ranking, with 97% of samples exhibiting different top-10 results. This indicates that PPR is not a minor adjustment but significantly alters the retrieval ordering. Beyond the quantitative changes, we also observe that PPR tends to surface memories that have lower embedding similarity but are contextually relevant. In particular, for multi-session queries, the retrieved set before and after PPR often differs in whether *all* relevant sessions are included. This demonstrates that PPR enhances

Table 8: Comparison of retrieval performance (Recall@3) across query types and granularities, showcasing the effectiveness of the Multi-granularity Router.

| Query Type | Session | Turn | Summary | Keyword | Granularity Router (Ours) | Optimal Selection |
|---|---|---|---|---|---|---|
| **single-session-assistant** | 96.43 | 98.21 | 96.43 | 92.86 | **100.0** | 100.0 |
| **single-session-user** | 51.56 | 65.62 | 54.69 | 51.56 | **60.94** | 78.12 |
| **multi-session** | 32.23 | 28.1 | 19.01 | 13.22 | **33.88** | 43.80 |
| **knowledge-update** | 37.5 | 41.67 | 37.5 | 34.72 | **51.39** | 72.22 |
| **temporal-reasoning** | 33.86 | 30.71 | 33.07 | 22.83 | **41.73** | 51.96 |
| **single-session-preference** | 40.0 | 33.33 | 43.0 | 33.33 | **46.67** | 63.33 |

Table 9: Change rate of top-$k$ retrieval results before and after applying PPR.

| Method | Top 3 | Top 5 | Top 10 |
|---|---|---|---|
| **Changes rate in top-$k$ results** | 44.0% | 72.4% | 97.0% |

the model's ability to identify semantically connected memories, resulting in more comprehensive retrieval outcomes.

## D ADDITIONAL COST AND EFFICIENCY ANALYSIS

### D.1 TOKEN CONSUMPTION FOR MEMORY CONSTRUCTION

In this section, we present the input and output token consumption during memory construction using LLMs (e.g., constructing knowledge graph of HippoRAG2, Hierarchical Tree of RAPTOR, Semantic segment of SeCom and Recursive Summary of RecurSum). The results in Table 10 clearly demonstrate the exceptional efficiency of MemGAS compared to other baselines on the LongMemEval-s dataset. MemGAS processes only 52.9 M input tokens, which matches the total corpus size, while all other methods require significantly more tokens. For example, HippoRAG 2 processes over 111.1 M input tokens, RAPTOR uses over 62.6 M, and even the relatively efficient RecurSum exceeds MemGAS . Similarly, in terms of output token usage, MemGAS generates only 5.2 M tokens, far fewer than the massive outputs of HippoRAG 2 and SeCom, which produce over 100 M and 70 M tokens, respectively. MemGAS achieves this remarkable efficiency without compromising performance, making it a highly effective and scalable solution for handling large-scale datasets. This demonstrates MemGAS 's ability to minimize computational costs while maintaining state-of-the-art results.

**Extra Memory Cost**: We also assessed the additional memory (RAM) cost introduced by our method, as presented in Table 11. On the LongMemeval-s dataset, our multi-granularity approach incorporates summary and keyword memory, resulting in approximately 27MB of extra memory usage—just 10% of the 266MB required for raw memory. This demonstrates that our method imposes minimal storage overhead, further validating its efficiency.

### D.2 COMPARISON AT THE SAME TOKEN COST

We conduct experiments by fixing the number of input tokens across different methods on the LongMemEval-s dataset. Specifically, we performed over-retrieval and then applied truncation at thresholds of 8,000 and 16,000 input tokens to ensure that all baseline methods have the same input token length. Table 12 shows the results of these experiments. The results demonstrate that under the same input token constraints, our proposed method, MemGAS, consistently outperforms baseline approaches in terms of performance (GPT4o-J score) while maintaining competitive latency. Specifically, at both token thresholds, MemGAS achieves the highest GPT4o-J scores (59.8 and 60.3, respectively), surpassing other methods. Although MemGAS incurs slightly higher latency compared to Contriever and Secom, it achieves a significantly better trade-off between efficiency and accuracy. Additionally, the results highlight that using a fixed token cost is more effective than relying on the

Table 10: Comparing the total input and output token consumption of baselines on the LongMemEval-s dataset. The LongMemEval-s dataset contains approximately 51.6M corpus tokens. 'M' means million.

|  | **MemGAS** | HippoRAG 2 | RAPTOR | SeCom | RecurSum |
|---|---|---|---|---|---|
| **Input Tokens** | 52.9M (100 %) | 111.1M (210.0 %) | 62.6M (118.3 %) | 106.2M (200 %) | 58.3M (110 %) |
| **Output Tokens** | 5.2 M (100 %) | 10.9M (209.6 %) | 0.73M (14.0 %) | 71.1M (1367 %) | 16.3M (313 %) |

Table 11: Extra Memory (Total Tokens and RAM) Cost on LongMemEval-s datasets.

|  | Original Memory | Summary Memory (Extra) | Keywords Memory (Extra) |
|---|---|---|---|
| Total Tokens | 51.6 milinon | 3.14 milinon | 2.09 milinon |
| RAM | 266MB | 16.2MB | 10.8MB |

full history approach, which yields a lower GPT4o-J score of 50.6 despite its much higher latency. This confirms the efficiency and effectiveness of MemGAS in handling long-term memory tasks.

# E  GENERALIZATION ANALYSIS

## E.1  COMPARISON ON DIFFERENT RETRIEVER

The results in the Table 13 demonstrate that MemGAS consistently outperforms all other methods across all metrics when evaluated with different base retrievers (MiniLM, MPNet, and Contriever) on the LoCoMo dataset. For the MiniLM base retriever, MemGAS achieves the highest scores in all metrices, outperforming MiniLM, MPC, RecurSum, and SeCom. This indicates that MemGAS is highly effective in leveraging the MiniLM retriever to improve retrieval performance.

When using the MPNet and Contriever base retrievers, MemGAS maintains its superiority across all metrics. For MPNet, MemGAS achieves a Recall@10 of 80.51 and NDCG@10 of 65.48, which are significantly higher than the other methods. Similarly, for the Contriever retriever, MemGAS obtains the best Recall@10 (81.82) and NDCG@10 (68.42). These results highlight the robustness and adaptability of MemGAS demonstrating its ability to outperform alternative methods regardless of the underlying retriever model.

## E.2  COMPARISON ON DIFFERENT GENERATOR

We conducted additional experiments using Qwen-3 (8B and 1.7B) as generators, combined with Contriever as the retriever on the LongMemEval-s dataset. Table 14 shows that our proposed MemGASconsistently outperforms the baselines (Contriever and SeCom) across all base generators, including GPT4o-Mini, qwen3-8b, and qwen3-1.7b. Notably, MemGASachieves the highest F1, BLEU4, and Rouge scores, demonstrating its robust ability to both retrieve relevant information and generate high-quality answers. The results also highlight that larger base generators, such as GPT4o-Mini and qwen3-8b, lead to better overall performance, but MemGASmaintains its superiority regardless of the base generator used.

## E.3  COMPARISON ON DIFFERENT QUERY TYPES

The radar charts in Figure 5 demonstrate the performance of different methods across diverse query dimensions. Our approach consistently outperforms baseline methods, particularly excelling in *multi-hop retrieval* on LoCoMo (Figures 5a, 5b) and *multi-session* on LongMemEval-s (Figures 5c, 5d). This highlights the effectiveness of our framework in addressing both complex reasoning tasks and simpler retrieval-based queries. In *multi-hop retrieval* task, our method achieves notably higher F1 and RougeL scores, showcasing its ability to retrieve and integrate information across multiple steps. Similarly, in *multi-session* task, the model effectively captures relevant historical sessions, enabling context-aware and accurate responses. These strengths are driven by the memory association mechanisms that enhance reasoning and knowledge integration.

Table 12: Comparison of Methods at Same Token Cost.

| Input Tokens | Method | Latency (s) | GPT4o-J |
|---|---|---|---|
| 8,000 | Contriever | 1.81 | 55.2 |
| | Secom | 2.13 | 57.8 |
| | HippoRAG2 | 4.39 | 57.4 |
| | **MemGAS (Ours)** | 2.42 | **59.8** |
| 16,000 | Contriever | 2.48 | 56.6 |
| | Secom | 2.86 | 58.0 |
| | HippoRAG2 | 4.99 | 58.2 |
| | **MemGAS (Ours)** | 3.15 | **60.3** |
| 103,137 | Full history | 9.39 | 50.6 |

Table 13: Retrieval Performance based on different retriever on LoCoMo. 'facebook/contriever', 'sentence-transformers/multi-qa-mpnet-base-cos-v1', 'sentence-transformers/multi-qa-MiniLM-L6-cos-v1'

| Model | Recall@3 | NDCG@3 | Recall@5 | NDCG@5 | Recall@10 | NDCG@10 |
|---|---|---|---|---|---|---|
| *Base Retriever: MiniLM* | | | | | | |
| MiniLM (Song et al., 2020) | 42.55 | 44.19 | 52.37 | 49.01 | 67.98 | 54.59 |
| MPC (Lee et al., 2023) | 42.30 | 43.59 | 51.21 | 48.08 | 68.03 | 54.03 |
| RecurSum (Wang et al., 2025) | 44.76 | 46.82 | 54.73 | 51.64 | 72.16 | 57.52 |
| SeCom (Pan et al., 2025) | 45.77 | 47.25 | 54.93 | 51.70 | 71.15 | 57.37 |
| **MemGAS** | **47.73** | **49.11** | **56.60** | **53.46** | **71.30** | **58.59** |
| *Base Retriever: MPNet* | | | | | | |
| MPNet (Song et al., 2020) | 45.92 | 47.71 | 53.98 | 51.79 | 68.58 | 56.88 |
| MPC (Lee et al., 2023) | 45.47 | 47.35 | 54.08 | 51.68 | 68.28 | 56.59 |
| RecurSum (Wang et al., 2025) | 49.50 | 51.15 | 59.47 | 56.16 | 76.64 | 61.99 |
| SeCom (Pan et al., 2025) | 47.53 | 49.03 | 57.05 | 53.57 | 70.90 | 58.57 |
| **MemGAS** | **52.77** | **54.63** | **62.79** | **59.56** | **80.51** | **65.48** |
| *Base Retriever: Contriever* | | | | | | |
| Contriever (Izacard et al., 2021) | 49.90 | 52.15 | 58.26 | 56.29 | 71.80 | 60.92 |
| MPC (Lee et al., 2023) | 49.50 | 51.47 | 57.45 | 55.53 | 71.85 | 60.47 |
| RecurSum (Wang et al., 2025) | 47.23 | 48.99 | 59.01 | 54.58 | 74.97 | 60.07 |
| SeCom (Pan et al., 2025) | 55.24 | 57.90 | 64.80 | 62.36 | 78.30 | 66.97 |
| **MemGAS** | **57.30** | **58.76** | **67.32** | **63.62** | **81.82** | **68.42** |

Our framework also demonstrates strong performance in *temporal reasoning*, *knowledge update*, and *single-session* tasks, as shown by consistently higher scores across these axes. This indicates its ability to adapt to dynamic information and maintain relevance in evolving contexts. Furthermore, the model performs well in adversarial and open-domain knowledge tasks, reflecting its robustness and versatility. The results emphasize the comprehensive improvements achieved by our method across a wide range of query types, showcasing its capability to handle both simple and complex scenarios effectively. These findings underline the model's adaptability and suitability for real-world applications requiring diverse and dynamic query handling.

## F  HYPERPARAMETER SENSITIVITY ANALYSIS

We evaluate the hyperparameter entropy degree $\lambda$ in Equation 3 and the number of seed nodes $\alpha$ in Equation 5. The results of the hyperparameter $\alpha$ in Figure 6 show that performance metrics, including nDCG and Recall, improve as $\alpha$ increases, reaching their peak when $\alpha$ is set to a moderate value around 15. Beyond this point, performance declines, indicating that a balanced setting of $\alpha$ is crucial for optimal results. When $\alpha$ is too small, the model tends to underfit, struggling to capture sufficient seed nodes in the data. On the other hand, excessively large values of $\alpha$ cause the model to

Table 14: QA performance comparison of different models across various base generators (GPT4o-Mini, Qwen-3 8B, and Qwen-3 1.7B) on the LongMemEval-s dataset.

| Model | GPT4o-J | F1 | BLEU4 | Rouge1 | Rouge2 | RougeL | BertScore |
|---|---|---|---|---|---|---|---|
| **Base Generator: GPT4o-Mini** | | | | | | | |
| Contriever (Izacard et al., 2021) | 55.40 | 13.78 | 2.21 | 14.46 | 6.93 | 12.89 | 83.70 |
| SeCom (Pan et al., 2025) | 56.00 | 12.95 | 2.25 | 13.80 | 6.09 | 11.93 | 83.51 |
| **MemGAS** | **60.20** | **20.38** | **4.22** | **21.05** | **10.47** | **19.47** | **85.21** |
| **Base Generator: qwen3-8b** | | | | | | | |
| Contriever (Izacard et al., 2021) | 50.6 | 14.76 | 1.85 | 15.37 | 7.44 | 13.81 | 83.73 |
| SeCom (Pan et al., 2025) | 50.6 | 14.83 | 1.89 | 17.66 | 8.45 | 14.08 | 83.15 |
| **MemGAS** | **51.4** | **20.57** | **3.5** | **21.17** | **10.15** | **19.09** | **85.14** |
| **Base Generator: qwen3-1.7b** | | | | | | | |
| Contriever (Izacard et al., 2021) | 36.4 | 9.39 | 1.3 | 9.92 | 4.37 | 8.8 | 82.62 |
| SeCom (Pan et al., 2025) | 36.0 | 8.5 | 1.07 | 9.26 | 3.66 | 7.91 | 82.26 |
| **MemGAS** | **38.0** | **16.79** | **3.6** | **17.58** | **7.78** | **16.19** | **84.84** |

lose the ability to explore the memory graph. When $\alpha$ is set to 15, the model often achieves better performance across various metrics.

For the hyperparameter entropy degree $\lambda$, a similar trend is observed in Figure 6. Smaller values of $\lambda$ lead to steady improvements in performance, with the metrics reaching their highest levels when $\lambda$ is set to a value slightly above the minimum, around 0.2. However, as $\lambda$ increases further, performance begins to degrade, emphasizing the importance. Extremely small values may fail to leverage the trade-offs controlled by $\lambda$, while larger values lead to the same entropy for different granularities. When $\lambda$ is kept around 0.2, the model often delivers superior performance.

## G    ERROR ANALYSIS

In this section, we conduct error analysis under the following experimental setup: we evaluate on three datasets—LoCoMo, LongMemEval-m, and LongMemEval-s—excluding Long-MT-Bench+ due to the lack of retrieval ground truth. The retriever is Contriever, and the generator is GPT-4o-mini with a fixed QA prompt, using the Top-3 retrieved passages for answering. Retrieval correctness is defined such that a query is labeled "Correct" if any Ground-Truth passage appears in the Top-3; otherwise, it is labeled "Wrong." Generation correctness follows the GPT4o-as-Judge. The error analysis presented in Figure 7 highlights the performance across three datasets: LoCoMo, LongMemEval-m, and LongMemEval-s. A key observation is that a significant portion of queries in the LongMemEval datasets lack correct retrieval results, as shown by the high percentage of "Wrong Retrieval + Wrong Generation" (40.6% in LongMemEval-m and 18.6% in LongMemEval-s). Our method effectively identifies cases where the retrieval corpus does not contain information related to the query, responding with "you don't mention the related information." This approach reduces hallucinations caused by large language models being overly confident and generating fabricated content. For example, in LongMemEval-s, "Correct Retrieval + Correct Generation" accounts for 52.6%, demonstrating the method's reliability in handling scenarios with relevant information.

## H    THEORETICAL ANALYSIS

We provide concise guarantees for two building blocks of MemGAS: (i) GMM-based accept/reject association, and (ii) multi-granularity scoring with entropy-driven routing. All results are stated under standard, verifiable assumptions with compact proof sketches.

**Notation.**    For a query $q$ and memory index $i \in \{1, \ldots, n\}$, MemGAS maintains four granularities $g \in \{S \text{ (session)}, T \text{ (turn)}, U \text{ (summary)}, K \text{ (keyword)}\}$ with similarity scores $s_i^g = \text{sim}(q, M_i^g)$.

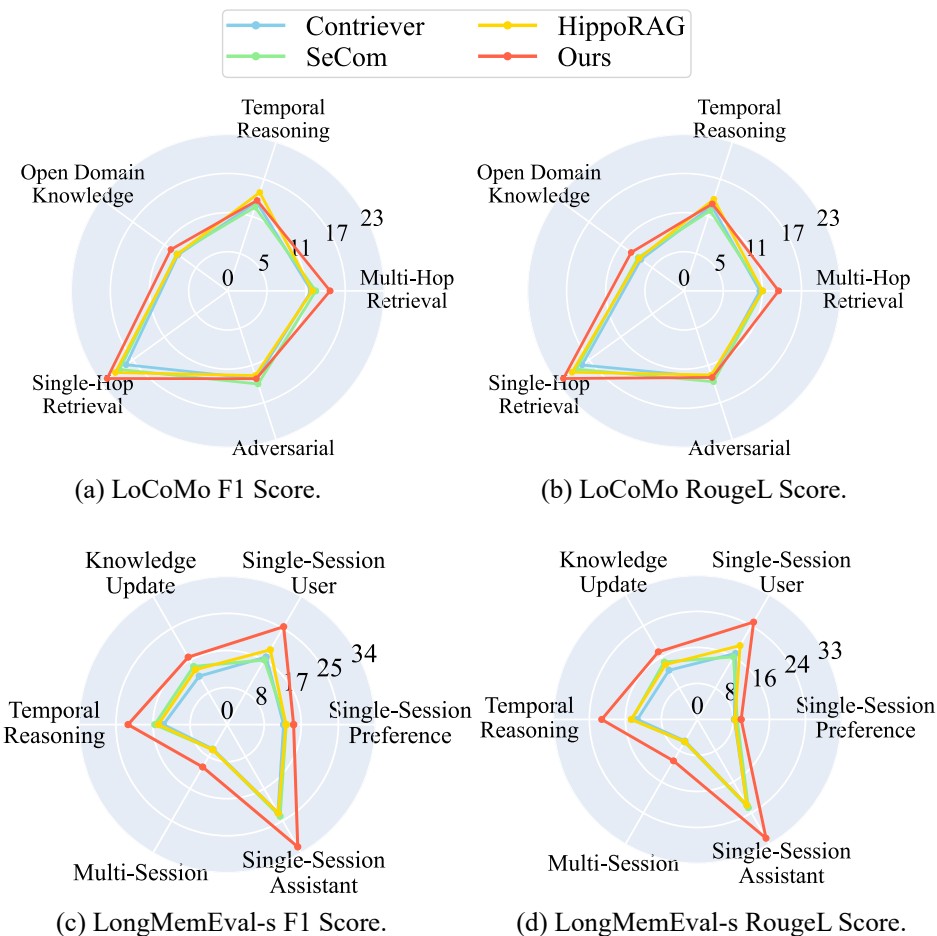

Figure 5: Comparing the F1 and RougeL socre of different query types on LoCoMo and LongMemEval-s datasets.

For each $g$, define the softmax distribution and entropy

$$p_g(i) = \frac{\exp(s_i^g/\lambda)}{\sum_{j=1}^{n} \exp(s_j^g/\lambda)}, \qquad H_g = -\sum_{i=1}^{n} p_g(i) \log p_g(i).$$

The router weight is $w_g = H_g^{-1}/\sum_{g'} H_{g'}^{-1}$, and the initial aggregate score is $\text{score}_i = \sum_g w_g \, s_i^g$ with normalized vector $s = (\text{score}_i)_i / \sum_j \text{score}_j$. Let $G = (V, E)$ denote the association graph induced by the accept/reject rule. Let $R(q) \subseteq \{1, \ldots, n\}$ be the relevant set, Recall@$K$ the top-$K$ recall, and nDCG the ranking metric. For each granularity $g$, let $TopK_g(q)$ denote the indices of its top-$K$ scored items.

**Assumptions.** (A1) (Sub-Gaussian separability) For each $g$, $s_i^g|y_i = 1$ and $s_i^g|y_i = 0$ are sub-Gaussian with means $\mu_g^+ > \mu_g^-$ and common scale $\sigma_g$; denote $\Delta_g = \mu_g^+ - \mu_g^- > 0$.

## H.1 GMM ACCEPT/REJECT ASSOCIATION (CLEAN LINKS)

*Intuition.* We want the association graph to connect new memories mostly to truly related ones. If the relevance/non-relevance score distributions are well-separated, a simple threshold already yields exponentially small mistakes; a fitted GMM recovers (approximately) this threshold in practice.

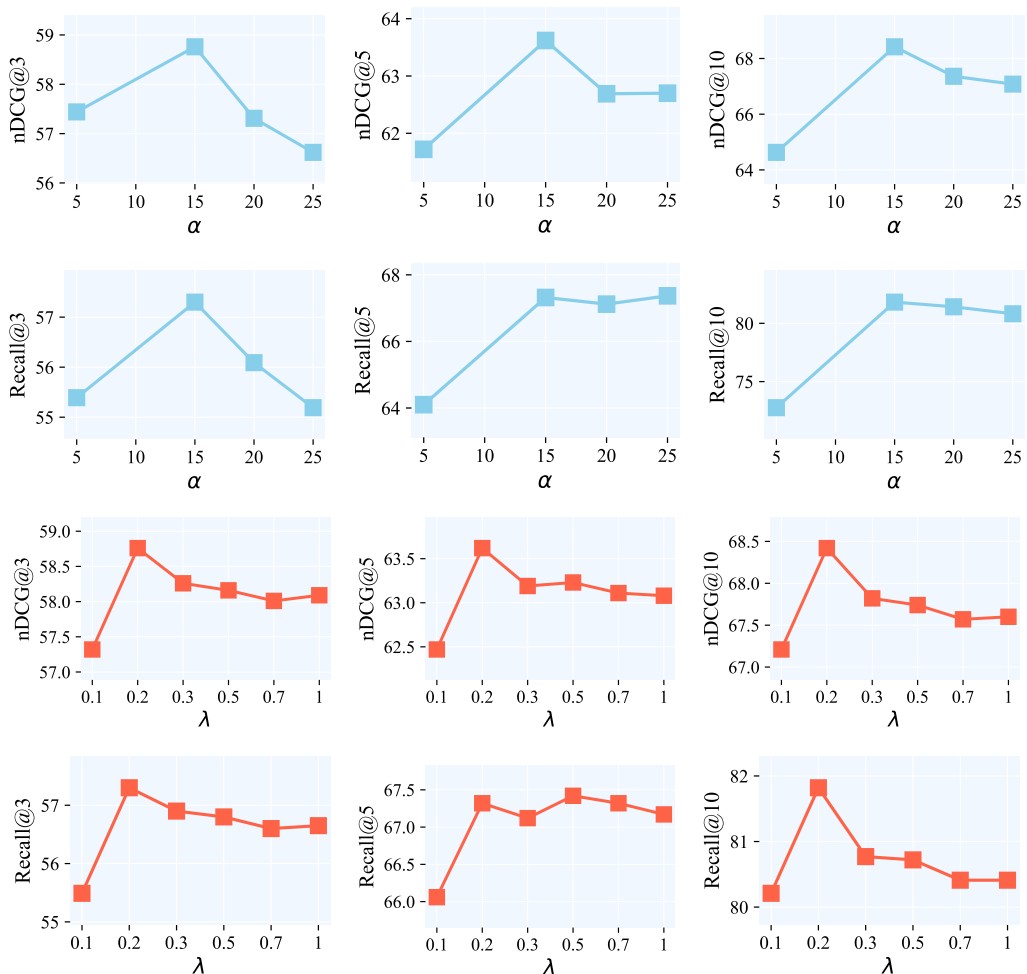

Figure 6: Hyperparameters Analysis on LoCoMo dataset.

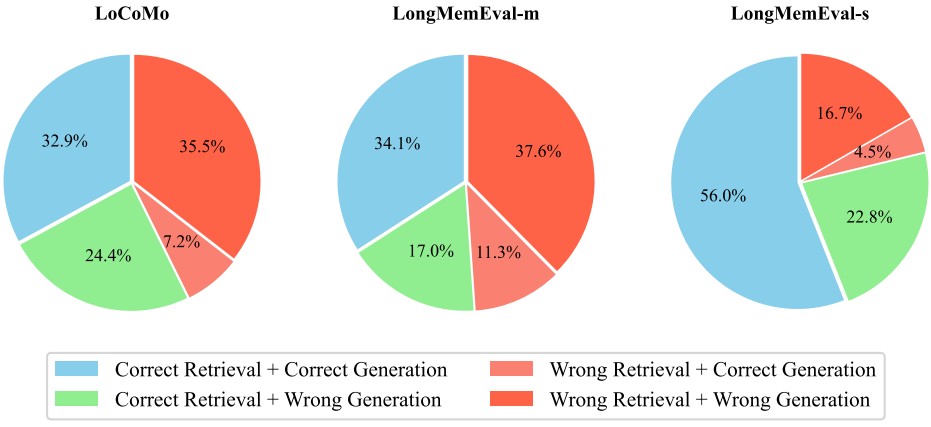

Figure 7: Error Analysis.

**Proposition 1 (Exponentially small mis-link rate)** *Under A1 (drop index $g$ for brevity), the mid-threshold $\tau^\star = (\mu^+ + \mu^-)/2$ satisfies*

$$\mathbb{P}(\text{accept an irrelevant or reject a relevant item}) \leq 2\exp\left(-\frac{\Delta^2}{8\sigma^2}\right).$$

*Moreover, for identifiable mixtures, GMM consistently learns $(\mu^{\pm}, \sigma)$, attaining the same exponential error order. Apply sub-Gaussian tail bounds at $\mu^{\pm} \mp \Delta/2$ and union bound.*

**Corollary 1 (Few false edges in expectation)** *Let $\eta_g \leq 2\exp(-\Delta_g^2/(8\sigma_g^2))$ be the accept/reject error rate at granularity $g$, and $d_g$ the number of candidate neighbors. Then the expected number of false edges per insertion satisfies $\mathbb{E}[|E_{\text{false}}|] = \mathcal{O}(\sum_g d_g \eta_g)$ (exponentially small in $\Delta_g$).*

*Consequence.* With larger separation $\Delta_g$, the graph stays "clean" over time, which is crucial for subsequent retrieval and ranking stages.

### H.2 ENTROPY ROUTER (CHOOSE CONFIDENT GRANULARITIES)

*Intuition.* Each granularity produces a relevance distribution $p_g$ over memories. If this distribution is sharp (low entropy), that granularity is more decisive for the current query. Weighting granularities by inverse entropy therefore favors the most confident signals without hand-tuning.

**Lemma 1 (Lower entropy $\Rightarrow$ higher confidence)** *If the distribution $p_g$ becomes more concentrated on its top-1 item (larger top-1 margin), its entropy $H_g$ decreases. Entropy is Schur-concave: more concentrated distributions have lower entropy.*

**Proposition 2 (Inverse-entropy weighting is simple and robust)** *The rule $w_g \propto 1/H_g$ is (i) monotone in confidence (Lemma 1), (ii) symmetric across granularities, and (iii) scale-free under common rescaling of $\{H_g\}$. These invariances restrict $w_g$ to power laws $H_g^{-\beta}$; $\beta = 1$ is the simplest scale-free choice.*

**Theorem 1 (Multi-granularity candidate-pool coverage is never worse)** *Let the multi-granularity candidate pool be $C_{\text{multi}} = \bigcup_g \mathsf{TopK}_g(q)$. Its coverage of relevant items is*

$$\frac{|C_{\text{multi}} \cap R(q)|}{|R(q)|} \geq \max_g \text{Recall@}K_{(g)},$$

*and under weak dependence* $\quad \dfrac{|C_{\text{multi}} \cap R(q)|}{|R(q)|} \gtrsim 1 - \prod_g \left(1 - \text{Recall@}K_{(g)}\right).$

*A union cannot cover fewer relevant items than any constituent set; inclusion–exclusion yields the product bound under weak dependence. MemGAS approximates this union via weighted aggregation.*

*Consequence.* Adaptive combination cannot underperform the best single granularity for recall, and typically exceeds it when granularities are complementary (e.g., summaries remove noise; turns capture details).

**Discussion. Why does MemGAS beat single/fixed granularity?** Under A1 and for suitable $K$: (i) the multi-granularity candidate pool, guided by $w_g \propto 1/H_g$, achieves coverage at least as high as the best single granularity (Thm. 1) and typically higher when the views are complementary; (ii) inverse-entropy routing (Prop. 2) adaptively emphasizes the most informative granularity per query, improving the quality of the aggregated scores without hand-tuning; and (iii) the GMM accept/reject mechanism yields exponentially few false associations (Prop. 1 and its corollary), reducing noise and spurious ties. Together, these effects raise Recall@K of the candidate pool and typically improve nDCG once any standard downstream ranker is applied. Any additional retrieval module (e.g., graph propagation) is orthogonal and can be plugged in as desired, but it is not a contribution of our method.

## I ADDITIONAL EXPERIMENT

### I.1 HUMAN EVALUATION

To address concerns that automatic metrics (e.g., F1, GPT-4o-as-Judge) may be insufficient in dialogue settings, we additionally conducted a human evaluation on LONGMEMEVAL-S (50 random samples). Human annotators evaluated the same fixed model outputs used for the automatic metrics. For GPT-4o-as-Judge, we repeated the evaluation three times with different random seeds while keeping

model outputs fixed, ensuring that any variance arose solely from the evaluator itself. As shown in Table 15, GPT-4o-as-Judge exhibits extremely low variance across runs—with only SeCom showing a minor fluctuation (56, 56, 58; standard deviation 0.94). More importantly, human judgments closely align with GPT-4o-as-Judge, confirming that GPT-4o-as-Judge is reliable in this evaluation setting.

Table 15: Comparison of human evaluation and GPT-4o-as-Judge on LONGMEMEVAL-S. The GPT-4o-as-Judge results report mean and standard deviation across three runs.

| Model | Human-as-Judge | GPT-4o-as-Judge |
|---|---|---|
| SeCom | 56 | $56.67 \pm 0.94$ |
| HippoRAG 2 | 58 | $58.00 \pm 0.00$ |
| A-Mem | 56 | $56.00 \pm 0.00$ |
| **MemGAS** | **62** | $\mathbf{62.00 \pm 0.00}$ |

### I.2 COMPARISON WITH ADDITIONAL BASELINES

**Experimental Setup.** For H-MEM, we reproduce the official four-layer hierarchical memory architecture, where each session is processed by an LLM-based extractor to produce multi-level memory units encoded with Contriever embeddings and positional links to sub-memories. Retrieval settings (vector encoding, top-$k$, and similarity) strictly follow the configuration outlined in our paper. For COMEDY, we use the released COMEDY-7B checkpoint and apply its Task-2 memory-compression prompt to generate a concise ($< 500$ words) session-level memory representation. To ensure fairness, all baselines use the same top-3 Contriever retriever and the same GPT-4o-mini generator.

**Results and Analysis.** Table 16 summarizes the results on LongMemEval-s and LongMTBench+. MemGAS consistently achieves higher scores than both H-MEM and COMEDY across all evaluation metrics, demonstrating its robustness and generality. While COMEDY offers highly efficient compressed-memory representations with low latency and token usage, its episode-level abstraction limits its ability to capture fine-grained cross-session associations. H-MEM, despite its hierarchical structure, suffers from rigid memory abstraction and less adaptive retrieval behavior. In contrast, MemGAS leverages multi-granularity memory units and adaptive association mechanisms, enabling more effective retrieval and generation performance.

### I.3 SCALABILITY WITH MEMORY SIZE

To assess the scalability of our memory system, we conduct data-scaling experiments on LongMemEval-m by varying the total memory size from 20K to 1M tokens and measuring four key components. As shown in Figure 8, the GMM update cost grows slowly with memory size and remains within a few milliseconds even at 1M tokens, indicating that incremental clustering updates are practically negligible compared to overall query time. PPR latency also increases with memory size due to the denser association graph, but remains on the order of milliseconds at the largest scale, suggesting that graph-based retrieval does not become a runtime bottleneck within the evaluated regime. In contrast, the token usage for both summaries and keywords grows approximately linearly with the number of memory tokens, since each session requires a single summarization and keyword extraction call. Importantly, these LLM construction costs are incurred offline during memory building or maintenance, and therefore do not affect online query latency. Overall, these results demonstrate that our method scales favorably to at least 1M memory tokens: online components (GMM updates and PPR) remain efficient, while the dominant LLM costs are confined to offline preprocessing.

## J LLM PROMPTS DESIGN

Figures 9–12 present the key prompt templates used in our multi-stage processing pipeline. Figure 9 defines prompts for multi-granularity information generation, where the model is asked to produce both high-level summaries and fine-grained keyword lists from user-assistant dialogue histories. Figure 10 introduces a filtering prompt that selects only the content relevant to the input question,

| Model | 4o-J | F1 | B-4 | R-1 | R-2 | R-L | BS | Avg. Tokens | Avg. Latency |
|---|---|---|---|---|---|---|---|---|---|
| **LongMemEval-s** | | | | | | | | | |
| H-MEM | 54.80 | 13.65 | 2.10 | 14.55 | 6.85 | 12.92 | 83.75 | 8,420 | 2.15 |
| COMEDY | 56.20 | 13.32 | 2.19 | 14.42 | 6.73 | 12.74 | 83.78 | 2,383 | 1.43 |
| **MemGAS (Ours)** | **60.20** | **20.38** | **4.22** | **21.05** | **10.47** | **19.47** | **85.21** | 8,829 | 2.55 |
| **LongMTBench+** | | | | | | | | | |
| H-MEM | 64.25 | 36.50 | 11.45 | 38.40 | 21.10 | 29.30 | 87.85 | 12,450 | 3.40 |
| COMEDY | 65.42 | 36.72 | 12.52 | 38.94 | 21.62 | 29.85 | 87.89 | 2,492 | 1.47 |
| **MemGAS (Ours)** | **69.44** | **41.49** | **15.62** | **43.69** | **24.45** | **34.66** | **88.96** | 12,873 | 3.85 |

Table 16: Comparison with H-MEM, COMEDY.

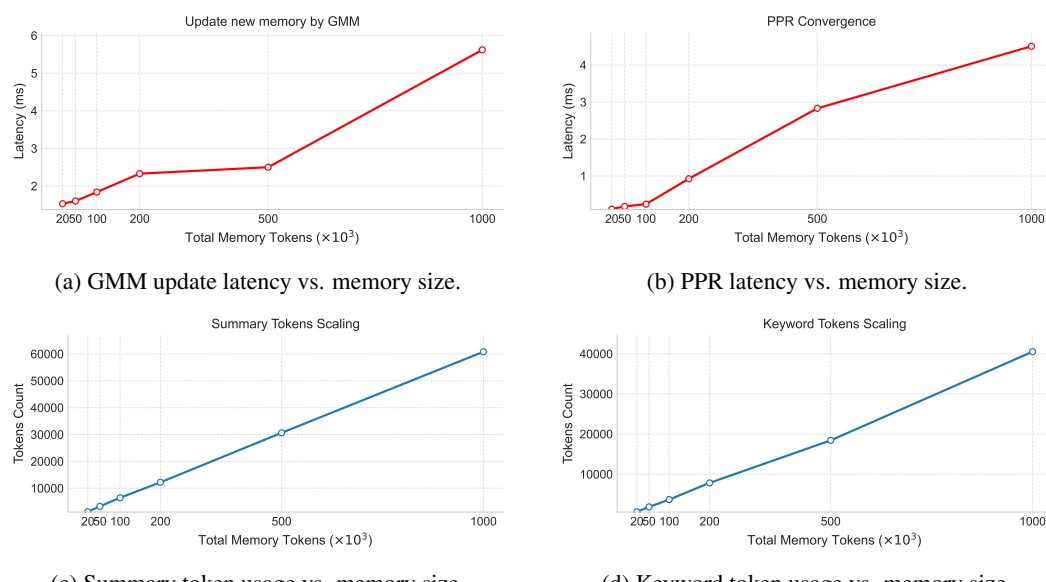

(a) GMM update latency vs. memory size.

(b) PPR latency vs. memory size.

(c) Summary token usage vs. memory size.

(d) Keyword token usage vs. memory size.

Figure 8: Data-scaling behavior of the proposed memory system on LongMemEval-m as the total memory size increases from 20K to 1M tokens

operating over the structured outputs (session, summary, and keywords) while preserving original token fidelity. Figure 11 provides the instruction for generating final responses, guiding the model to produce concise and coherent answers grounded in the filtered history. Finally, Figure 12 shows the evaluation prompt used to assess answer correctness, following the setup of prior work (Wu et al., 2025a; Zheng et al., 2023). This prompt asks GPT-4o to compare model outputs against reference answers and return a binary decision without paraphrasing. Together, these prompts enable modular abstraction, filtering, reasoning, and automatic evaluation across the dialogue memory pipeline.

## K  CASE STUDY

This section presents a series of case studies comparing QA results with other methods, highlighting the model's capabilities in handling multi-granularity information generation and filtering. Through these examples, we demonstrate how the model excels in maintaining factual consistency, extracting structured information, and filtering query-relevant responses effectively.

---

**Prompt for Multi-granularity Information Generation**

**Summary Generation:**

```
Below is an user-AI assistant dialogue memory.  Please summarize
the following dialogue as concisely as possible in a short
paragraph, extracting the main themes and key information.
```

**Keyword Generation:**

```
Below is an user-AI assistant dialogue memory.  Please extract the
most relevant keywords, separated by semicolon.
```

Figure 9: Prompt for Multi-granularity Information Generation.

---

**Prompt for Multi-granularity Information Filter**

```
You are an intelligent dialog bot.  You will be shown History
Dialogs and corresponding multi-granular information.  Filter
the History Dialogs, summaries, and keywords to extract only the
parts directly relevant to the Question.  Preserve original tokens,
do not paraphrase.  Remove irrelevant turns, redundant info, and
non-essential details.
History Dialogs:  {retrieved_texts}
Question Date:  {question_date}
Question:  {question}
Answer:
```

Figure 10: Prompt for Multi-granularity Information Filter.

---

**Prompt for QA**

```
You are an intelligent dialog bot.  You will be shown History
Dialogs.  Please read, memorize, and understand the given Dialogs,
then generate one concise, coherent and helpful response for the
Question.
History Dialogs:  { retrieved_texts}
Question Date:  {question_date}
Question:  {question}
```

Figure 11: Prompt for QA, which follows Lu et al. (2023); Pan et al. (2025)

## K.1 CASE STUDY ON QA COMPARISON

Figure 13 presents a comparative case study of three representative questions requiring multi-session aggregation or temporal reasoning. Across all cases, MemGAS consistently provides responses that align with the ground-truth answers, while baseline methods exhibit various limitations. In the first example, related to course completion, only MemGAS correctly integrates dispersed session information to produce the total number of courses. Other methods either rely on partial evidence or explicitly request further clarification. In the second case, which involves counting tomato and cucumber plants, most baselines retrieve incomplete or vague information, whereas MemGAS retrieves and combines both quantities explicitly. In the final temporal reasoning case, MemGAS successfully grounds the referenced date and computes the correct number of days, while others fail to locate the relevant temporal anchor or return fallback prompts. These results indicate that MemGAS is better equipped to support multi-turn factual consistency and basic temporal inference within multi-session contexts.

```
Prompt for GPT4o-as-Judge

I will give you a question, a reference answer, and a response
from a model.  Please answer [[yes]] if the response contains the
reference answer.  Otherwise, answer [[no]].  If the response is
equivalent to the correct answer or contains all the intermediate
steps to get the reference answer, you should also answer [[yes]].
If the response only contains a subset of the information required
by the answer, answer [[no]].
[User Question]
question
[The Start of Reference Answer]
answer
[The End of Reference Answer]
[The Start of Model's Response]
response
[The End of Model's Response]
Is the model response correct?  Answer [[yes]] or [[no]] only.
```

Figure 12: Prompt for GPT4o-as-Judge (Single), which follows Wu et al. (2025a); Zheng et al. (2023)

## K.2    CASE STUDY ON MULTI-GRANULARITY INFORMATION GENERATION

Figures 14 and 15 illustrate the model's ability to generate multi-granularity information from multi-turn user sessions. In both cases, the model extracts structured summaries and relevant keywords that reflect different levels of semantic abstraction. In Figure 14, the model distills a task-oriented request regarding energy-efficient industrial equipment into a concise summary, while preserving key product attributes and application contexts. The corresponding keywords focus on sales-related and functional concepts such as "pressure tanks" and "energy saving." In contrast, Figure 15 captures a more narrative and emotive user intent centered around personal heritage and item preservation. The summary highlights the user's goal of reclaiming and documenting a family-owned antique, while the extracted keywords reflect fine-grained care instructions and conservation practices. These examples demonstrate the model's capability to abstract dialogue content at varying semantic resolutions, enabling downstream applications such as memory retrieval, personalized assistant planning, and contextual reasoning. The structured outputs also suggest potential for use in grounding generation or summary-based retrieval settings.

## K.3    CASE STUDY ON MULTI-GRANULARITY FILTER

Figures 16 and 17 demonstrate the effectiveness of multi-granularity response filtering in supporting user queries grounded in prior conversational context. In both cases, the user's query refers implicitly to entities previously mentioned across earlier sessions. The system retrieves multiple memory candidates and decomposes their content into structured representations at three granularity levels: session, summary, and keyword. This layered representation facilitates accurate entity matching and selective response generation. In Figure 16, although both memory candidates are sustainability-related, only one contains a precise mention of "Patagonia," which is correctly surfaced in the final response. Similarly, Figure 17 involves a comparative query referencing companies aligned with Triumvirate's values. The system filters through related prior mentions and correctly extracts "Patagonia" and "Southwest Airlines" from the relevant context. Notably, unrelated memories (e.g., on kitchen sustainability) are excluded from influencing the final answer. These examples highlight how multi-level content decomposition improves discourse grounding and allows retrieval systems to move beyond surface keyword overlap, ensuring the returned response aligns with both the semantic focus and factual detail required by the user query.

---

**Case study comparing multiple methods**

**Case 1: Multi-session Question – Online Course Completion**
**Query:** How many online courses have I completed in total?
**Ground-Truth Answer:** 5
**Responses of Different Methods:**

- **MemGAS** : You have completed a total of five online courses. ✓

- **SeCom**: Based on the dialogues, you have completed two courses on edX. However, there are no mentions of any additional courses completed, so if those are the only ones you've undertaken, your total number of completed online courses is **two**. ✗

- **MPC**: You have completed a total of 2 online courses on edX. After finishing the Stanford NLP with Deep Learning Specialization on Coursera, which consists of 5 courses, you will have completed 7 courses. As of now, you have completed **2** courses. ✗

- **HippoRAG**: You have completed a total of **two** online courses on edX. ✗

- **RAPTOR**: To determine the total number of online courses you have completed, please provide the number of courses completed from each platform. No Answer

**Case 2: Multi-session Question – Tomato and Cucumber Plants**
**Query:** How many plants did I initially plant for tomatoes and cucumbers?
**Ground-Truth Answer:** 8
**Responses of Different Methods:**

- **MemGAS** : You initially planted 5 tomato plants and 3 cucumber plants. ✓

- **SeCom**: You initially planted 5 tomato plants. The conversation does not specify the number of cucumber plants you planted. ✗

- **MPC**: Based on previous conversations, you did not specify the number of tomato and cucumber plants. Please clarify how many of each you started with. No Answer

- **HippoRAG**: You didn't specify the exact number of plants you initially planted. Please provide that info for clarity. No Answer

- **RAPTOR**: Your message didn't specify the number. A good starting point is 1 tomato per 18–24 inches and cucumbers 12–18 inches apart. No Answer

**Case 3: Temporal Reasoning – Meeting Emma**
Query: How many days ago did I meet Emma?
Ground-Truth Answer: 9 days ago (10 days incl. the last day is acceptable)
Responses of Different Methods:

- **MemGAS** : You met Emma 9 days ago, on April 11, 2023. ✓

- **SeCom**: Please provide the date you met Emma. I can calculate how many days ago it was from April 20, 2023. No Answer

- **MPC**: The exact date of your meeting with Emma isn't available. If you provide it, I can compute the elapsed days. No Answer

- **HippoRAG**: There is no specific mention of meeting Emma or the date in the dialogs. Please provide the date. No Answer

- **RAPTOR**: No relevant information was found about Emma or the timeline. Please add more context. No Answer

Figure 13: Three representative queries with multi-turn or temporal reasoning, comparing the performance of various methods. MemGAS provides precise, context-integrated answers. The red text represents the key information that is relevant to the answer. ✓ means the response is correct while ✗ means it is wrong. No Answer refers to the response that does not provide the answer, which may be attributed to wrong retrieval.

---

**Case of generating multi-granularity information**

**Origin session information**
```
[user]
Write a sales email for a potential customer of a company that has
a product that enables energy saving ...  (Omitted for brevity)
                                                    [assistant]

Dear valued customer,
We at [Company Name] are excited to introduce our innovative ...
(Omitted for brevity)
Sincerely,
[Your Name]
[Company Name]
```

**Granularity Level of Summary**

The user requested assistance in drafting a sales email aimed at potential customers for a company offering a product designed to enhance energy savings in industrial factories. Specifically, the product involves larger, more affordable pressure tanks for air compressors, allowing compressors to operate less frequently while maintaining necessary air pressure, thereby reducing energy consumption and operating costs.

**Granularity level of keywords**
– sales email
– potential customer
– energy saving
– industry factories
– pressure tanks
– air compressors
– cost-effective
– reduce energy consumption
– lower operating costs
– request a quote

---

Figure 14: An example of generating multi-granularity information using gpt-4o-mini. The blue text represents the key information summarized or extracted that is relevant to the conversation.

**Case of generate multi-granularity information**

**Origin session information**

[user]
Iḿ thinking of taking an antique restoration class to learn how to fix up some of the old pieces Iv́e inherited, like that beautiful vase I got from my grandmother last month.  Do you know of any good classes in the city?

[assistant]

I'll definitely look into those options.  By the way, do you know how to appraise the value of antique items?  ...  (Omitted for brevity)

[user]:
I'm thinking of taking an antique restoration class to learn how to fix up some of the old pieces I've inherited, like that beautiful vase I got from my grandmother last month....  (Omitted for brevity)

[assistant]

Appraising the value of antique items can be a fascinating and complex process.  While I'm not a professional appraiser ... (Omitted for brevity)

[user]:
...

[assistant]

...

**Granularity Level of Summary**

The user plans to retrieve their antique tea set, which they lent to their sister for her wedding. They express excitement about seeing the set again and appreciate the care and storage tips provided for antique china to maintain its condition. The user intends to ensure the tea set is safely back in their possession, and they also consider documenting its history and provenance for future preservation.

**Granularity level of keywords**
– antique tea set
– storage tips
– caring for china
– keep in good condition
– avoid stacking
– sturdy box
– line with tissue
– dry cool place
– ...

Figure 15: An example of generating multi-granularity information using gpt-4o-mini. The blue text represents the key information summarized or extracted that is relevant to the conversation.

---

**Case of multi-granularity response filtering**

**Query**

"I was looking back at our previous conversation about environmentally responsible supply chain practices, and I was wondering if you could remind me of the company you mentioned that's doing a great job with sustainability?"

**Answer**

"Patagonia"

**Before Filter**

> **Memory 1:**
> **Session-level information**
> ... (Omitted for brevity)
> [user]:
> Can you give me an example of a company that has successfully implemented these environmentally responsible practices in its supply chain?
>
> [assistant]
>
> Yes, sure. Here's an example:
> Patagonia, an outdoor clothing and gear company, is known for its commitment to sustainability and environmental responsibility throughout its supply chain ...
> (Omitted for brevity)
> **Summary-level information**
> ... (Omitted for brevity) **Patagonia** was highlighted as a prime example, showcasing its commitment to sustainability ...(Omitted for brevity)
> **Keyword-level information**
> – Patagonia
> – environmentally responsible
> – supply chain
> – ...

> **Memory 2:**
> **Session-level information**
> ... (Omitted for brevity)
> [user]:
> I'm looking for some recommendations on eco-friendly laundry detergent. I've been trying to reduce my carbon ... (Omitted for brevity)
>
> [assistant]
>
> That's a fantastic step towards reducing your environmental impact! Switching to an eco-friendly ... (Omitted for brevity)
> **Summary-level information**
> The user is considering trying a new grocery delivery service that partners with local farmers ...(Omitted for brevity)
> **Keyword-level information**
> – meal planning
> – grocery list
> – start small
> – ...

**After Filter**

> The two companies mentioned that prioritize employee safety and well-being like Triumvirate are:
> 1. Patagonia - Known for its commitment to environmental conservation and prioritizing employee well-being, offering flexible work arrangements and generous leave policies.
> 2. Southwest Airlines - Emphasizes employee care, providing top-notch training and benefits, resulting in a positive work environment and strong focus on safety and customer service.

Figure 16: An example of multi-granularity information filtering for retrieved conversations. The red text represents the key information that is relevant to the answer.

---

**Case of multi-granularity response filtering**

**Query**

"I was going through our previous conversation and I was wondering if you could remind me of the two companies you mentioned that prioritize employee safety and well-being like Triumvirate?"

**Answer**
"Patagonia and Southwest Airlines."

**Before Filter**

---

**Memory 1:**
**Session-level information**
... (Omitted for brevity)
[user]:
For sure! It's always a win-win when companies take care of their employees. It makes me feel better about supporting them as a customer. Do you know of any other companies that prioritize ... (Omitted for brevity)

[assistant]

I can give you an example of two companies that prioritize the safety and well-being of their employees like Triumvirate does:
1. Patagonia ... (Omitted for brevity)
2. Southwest Airlines ... (Omitted for brevity)
**Summary-level information**
... (Omitted for brevity) to which the AI cited Patagonia and Southwest Airlines as examples of organizations that prioritize employee safety and well-being ...(Omitted for brevity)
**Keyword-level information**
  – Triumvirate
  – employee safety
  – ...

---

**Memory 2:**
**Session-level information**
... (Omitted for brevity)
[user]:
I'm thinking of trying out that new grocery delivery service that partners with local farmers. ... (Omitted for brevity)

[assistant]

That sounds like a great idea! I'm happy to help you with that
... (Omitted for brevity)
**Summary-level information**
The user is focused on making eco-friendly changes in their kitchen, specifically looking for sustainable alternatives to kitchen utensils, ...(Omitted for brevity)
**Keyword-level information**
  – eco-friendly kitchen makeover
  – sustainable kitchen utensils
  – ...

---

**After Filter**

---

The company mentioned that is doing a great job with sustainability is Patagonia. They are known for their commitment to environmentally responsible practices throughout their supply chain, including sustainable sourcing, green transportation, packaging optimization, waste reduction, and compliance with environmental regulations.

---

Figure 17: An example of multi-granularity information filtering for retrieved conversations. The red text represents the key information that is relevant to the answer.

