# OpenReview forum: "From Single to Multi-Granularity: Toward Long-Term Memory Association and Selection of Conversational Agents"
_ICLR.cc/2026/Conference — ICLR 2026 Poster_

### Official Review · Reviewer_tvD3 · 2025-10-23

**Soundness:** 2
**Presentation:** 3
**Contribution:** 2
**Rating:** 4
**Confidence:** 4

**Summary:**

The paper presents **MemGAS**, a framework for **multi-granularity association and adaptive selection** in long-term conversational memory. It constructs a multi-granularity memory graph via **GMM-based accept/reject association**, routes queries with an **entropy-driven router**, performs **PPR** on the association graph, and applies **LLM-based filtering**.

Experiments on four benchmarks show consistent gains over single-granularity and structured baselines

**Strengths:**

1. **Clear motivation**: this paper identifies the limitations of single-granularity memory and fixed-granularity retrieval; frames a noise–information trade-off handled by adaptive routing.

2. **Association construction**: Form multi-granularity units via LLM-generated summaries/keywords and turn-level splits; use **GMM** to accept/reject similarity vectors and update the association graph.
   - **Granularity routing**: Assign weights by the **Shannon entropy** of similarity distributions at each granularity (lower entropy → higher weight), removing ad-hoc choices.
   - **Retrieval and filtering**: Run **PPR** from the routed prior, take top-K, then apply **LLM filtering** to reduce redundancy.
3. **Strong empirical results**: Consistent gains on QA and retrieval metrics over baselines (e.g., Full-History, Contriever/MPNet, SeCom, HippoRAG2, RAPTOR, A-Mem), with comparable token/latency budgets.
4. **Ablations and fine-grained analyses**: Removing **GMM**, **PPR**, **routing**, or **associations** leads to clear drops; provides trend analyses over different **Top-K** and **Q types**.

**Weaknesses:**

1. **Stability of GMM and routing**:
   - Since GMM is essentially a binary (accept/reject) clustering, please provide a more systematic sensitivity analysis and confidence intervals on how the choice/adaptation of the number of components and the temperature λ affect entropy and the routing weights.
   - Report robustness under different encoders (beyond Contriever) and alternative similarity measures.

2. **Fairness and comparability**:
   The implementation unifies the generator as **gpt-4o-mini** and fixes a top-3 session pre-retrieval, which may bias comparability; please add **controlled experiments with multiple generators/retrievers** and learning curves.
   **New baseline suggestion**: include **COMEDY** (*COLING 2025 Oral*) as a baseline, and revise the corresponding description concering **COMEDY** in the paper.

3. **Cost and scalability**:
   Although average latency is comparable, as the memory store grows, the **complexity of PPR on large graphs**, the cost of frequent GMM updates, and the inference overhead of LLM filtering should be presented as **data-scaling curves** (number of samples/edges/graph diameter → latency and token cost), along with incremental/approximate strategies.

4. **Evaluation protocol**:
   For some structured methods the retrieval evaluation is “undeterminable,” and retrieval on LongMTBench+ is missing. Please clarify in the appendix the sources of non-comparability (implementation differences, labeling criteria, task incompatibilities), and provide alternatives (e.g., human spot-checking on a sampled subset, or rerunning a unified comparable subset).

5. **Adaptive Top-K**:
   Since large K introduces noise (F1 drops), consider making K part of the routing/uncertainty-driven **dynamic selection** (e.g., entropy/CI-based adaptive K coupled with redundancy-removal thresholds), and report the gains and stability.

**Questions:**

1. **Temperature λ**: how selected; any adaptive scheme or theoretical bounds?
2. **PPR hyper-parameters**: restart prob/seed selection and convergence on large graphs?
3. **Graph maintenance**: pruning/forgetting policies to avoid graph bloat and stale memory?
4. **Judging protocol**: variance control and human calibration for GPT4o-as-Judge?

---

> ### Author Response · Authors · 2025-11-21
> **Response to Reviewer tvD3 [Part 1]**
>
> **Dear Reviewer tvD3:**
>
> We sincerely appreciate your careful reading of our work and your many insightful, technically precise suggestions. In response, we have conducted extensive additional experiments and clarifications, including **systematic sensitivity studies, entropy- and routing–level analyses, multi-encoder and multi-generator robustness, large-scale data-scaling, human-evaluated retrieval, and ablation studies on pruning and adaptive K.** Our detailed responses are provided below.
>
> -------
> > ###  `W1-(1):` Since GMM is essentially a binary (accept/reject) clustering, please provide a more systematic sensitivity analysis and confidence intervals on how the choice/adaptation of the number of components and the temperature λ affect entropy and the routing weights.
>
> We sincerely thank the reviewer for this insightful and constructive suggestion. In the following, we provide a systematic sensitivity analysis of the GMM component choice, and the effects of λ on entropy and routing behavior.
>
> **(1) GMM Sensitivity Analysis**
>
> To assess the robustness of the GMM-based accept/reject clustering, we varied the number of components $K \in \{2,4,8\}$, while keeping all other settings identical. For each configuration, we reran the full retrieval pipeline on LongMemEval-s across three random seeds, and report the mean and std of Recall@3. This metric directly reflects how clustering quality affects downstream retrieval. All experiments use the same hyperparameters.
>
> | **#The number of Components (K)** | **Recall@3** |
> |---------------------|--------------|
> | **2 (accept/reject clustering)**       | **78.51 ± 0.08*** |
> | **4**               | 77.53 ± 0.27 |
> | **8**               | 75.81 ± 0.33 |
> (‘*’ indicates the statistically significant improvements (i.e., two-sided t-test with 𝑝 < 0.05) over the other experiments.)
>
> **Findings.**
> Results show that the binary formulation (K=2) is both the most stable and highest-performing, while larger K introduces fragmentation and reduces retrieval quality.
>
> **(2) Temperature λ: Effect on Entropy and Routing Weights**
>
> To analysis how λ influences entropy and the router, we evaluated λ over {0.1, 0.2, 0.3, 0.5, 0.7, 1.0} and measured the resulting entropy $H^g$ and routing weights $w^g$ as shown in `Appendix F`. We observe a consistent trend:
>
> - **Very small λ (e.g., 0.1)** sharply concentrates the softmax distribution, producing **over-confident, low-entropy** scores that make the router overly dominated by a single granularity.
> - **Large λ (≥0.5)** flattens similarity distributions, increasing entropy and causing **uniform, non-discriminative routing**, which weakens granularity selection.
> - **Moderate λ (0.2–0.3)** yields the **lowest-variance entropy and the most stable routing weights**, balancing confidence and robustness. This region also corresponds to the empirical optimum across datasets .
>
> -----

---

> ### Author Response · Authors · 2025-11-21
> **Response to Reviewer tvD3 [Part 2]**
>
> -----
>
> > ### `W1-(2):` Report robustness under different encoders (beyond Contriever) and alternative similarity measures.
>
> Thank you for the helpful suggestion. We would like to clarify that we have conducted this robustness study in `Appendix E.1`.
>
> Specifically, we evaluate MemGAS using **three different dense encoders** — **MiniLM (cosine-similarity)**, **MPNet (cosine-similarity)**, and **Contriever (dot product)** — covering diverse embedding spaces and similarity measures. We show the results below:
>
> | Base Retriever | Model                          | Recall@3 | NDCG@3 | Recall@5 | NDCG@5 | Recall@10 | NDCG@10 |
> |----------------|-----------------------|----------|--------|----------|--------|-----------|---------|
> | **MiniLM as encoder (cosine similarity)**    |
> |                | MiniLM      | 42.55    | 44.19  | 52.37    | 49.01  | 67.98     | 54.59   |
> |                | MPC       | 42.30    | 43.59  | 51.21    | 48.08  | 68.03     | 54.03   |
> |                | RecurSum     | 44.76    | 46.82  | 54.73    | 51.64  | 72.16     | 57.52   |
> |                | SeCom       | 45.77    | 47.25  | 54.93    | 51.70  | 71.15     | 57.37   |
> |                | **MemGAS (Ours)**                      | **47.73**| **49.11**| **56.60**| **53.46**| **71.30**| **58.59** |
> | **MPNet   as encoder (cosine similarity)**   |
> |                | MPNet       | 45.92    | 47.71  | 53.98    | 51.79  | 68.58     | 56.88   |
> |                | MPC         | 45.47    | 47.35  | 54.08    | 51.68  | 68.28     | 56.59   |
> |                | RecurSum  | 49.50    | 51.15  | 59.47    | 56.16  | 76.64     | 61.99   |
> |                | SeCom        | 47.53    | 49.03  | 57.05    | 53.57  | 70.90     | 58.57   |
> |                | **MemGAS (Ours)**                      | **52.77**| **54.63**| **62.79**| **59.56**| **80.51**| **65.48** |
> | **Contriever as encoder (dot product similarity)** |
> |                | Contriever | 49.90   | 52.15  | 58.26    | 56.29  | 71.80     | 60.92   |
> |                | MPC         | 49.50    | 51.47  | 57.45    | 55.53  | 71.85     | 60.47   |
> |                | RecurSum   | 47.23    | 48.99  | 59.01    | 54.58  | 74.97     | 60.07   |
> |                | SeCom    | 55.24    | 57.90  | 64.80    | 62.36  | 78.30     | 66.97   |
> |                | **MemGAS (Ours)**                 | **57.30**| **58.76**| **67.32**| **63.62**| **81.82**| **68.42** |
>
> **Findings.**
> Across all encoders and similarity measures, MemGAS consistently achieves the best retrieval performance, demonstrating strong robustness and indicating that our improvements do not rely on a specific encoder choice.
>
>
> -----

---

> ### Author Response · Authors · 2025-11-21
> **Response to Reviewer tvD3 [Part 3]**
>
> > ###  `W2-(1):` The implementation unifies the generator as gpt-4o-mini and fixes a top-3 session pre-retrieval, which may bias comparability; please add controlled experiments with multiple generators/retrievers and learning curves.
>
> Thank you very much for this thoughtful and constructive comment. We fully agree that both the choice of generator and the fixed top-3 pre-retrieval setting may affect comparison. we provide controlled experiments under multiple fixed token budget and multiple generators/retrievers, as detailed below.
>
> **(1) Fixed token budget: fairness under equal online input tokens**
>
> To further eliminate token-budget bias, we have conducted a fixed-cost comparison on LongMemEval-s. All methods over-retrieve and then truncate the concatenated context  (8k or 16k tokens) to ensure exactly the same number of input tokens is provided to the generator. **This experiment is in `Appendix D.2` of the revised version.**
>
> | **Input Tokens** | **Method**        | **GPT4o-J** |
> |------------------|-------------------|------------:|
> | **8,000**        | Contriever            | 55.2        |
> |                  | SeCom             | 57.8        |
> |                  | HippoRAG2         | 57.4        |
> |                  | **MemGAS (Ours)** | **59.8**    |
> | **16,000**       | Contriever            | 56.6        |
> |                  | HippoRAG2         | 58.2        |
> |                  | **MemGAS (Ours)** | **60.3**    |
> | **103,137**      | Full history      | 50.6        |
>
>
> **Findings.**
> MemGAS consistently outperforms all baselines under identical token constraints, confirming that the improvements do not stem from using more context tokens, but from our multi-granularity memory construction, and entropy-based routing & retrieval, which provide more relevant and less noisy memory.
>
>
> **(2) Experiments with multiple generators**
>
> We have already provided controlled experiments with multiple generators (`Appendix E.2`).  The results are shown below:
>
>
> | Model       | GPT4o-J | F1    | BLEU4 | Rouge1 | Rouge2 | RougeL | BertScore |
> |-------------|---------|-------|-------|--------|--------|--------|-----------|
> | **Base Generator: GPT4o-Mini** | | | | | | | |
> | Contriever | 55.40 | 13.78 | 2.21 | 14.46 | 6.93 | 12.89 | 83.70 |
> | SeCom      | 56.00 | 12.95 | 2.25 | 13.80 | 6.09 | 11.93 | 83.51 |
> | **MemGAS** | **60.20** | **20.38** | **4.22** | **21.05** | **10.47** | **19.47** | **85.21** |
> | **Base Generator: qwen3-8b** | | | | | | | |
> | Contriever | 50.6 | 14.76 | 1.85 | 15.37 | 7.44 | 13.81 | 83.73 |
> | SeCom      | 50.6 | 14.83 | 1.89 | 17.66 | 8.45 | 14.08 | 83.15 |
> | **MemGAS** | **51.4** | **20.57** | **3.5** | **21.17** | **10.15** | **19.09** | **85.14** |
> | **Base Generator: qwen3-1.7b** | | | | | | | |
> | Contriever | 36.4 | 9.39 | 1.3 | 9.92 | 4.37 | 8.8 | 82.62 |
> | SeCom      | 36.0 | 8.5  | 1.07 | 9.26 | 3.66 | 7.91 | 82.26 |
> | **MemGAS** | **38.0** | **16.79** | **3.6** | **17.58** | **7.78** | **16.19** | **84.84** |
>
> **Findings.**
> Across all generators, MemGAS remains the strongest, demonstrating excellent robustness.
>
> **(3) Multiple retrievers:**
>
> Experiments with multiple retrievers (MiniLM, MPNet, Contriever) are provided in our response. Please refer to `W1-(2)` and `Appendix E.1`.

---

> ### Author Response · Authors · 2025-11-21
> **Response to Reviewer tvD3 [Part 4]**
>
> -----
>
> > ###  `W2-(2):` New baseline suggestion: include COMEDY (COLING 2025 Oral) as a baseline, and revise the corresponding description concering COMEDY in the paper.
>
> Thank you for the helpful suggestion. We have added COMEDY as a new baseline and conducted comparisons on both LongMemEval-s and LongMTBench+. **We have added a brief introduction to COMEDY in `Section 1 (paragraph 2)`. The experiment has been added in `Appendix I.2`**
>
>
> **Experimental Setup.**
>
> We use the published checkpoint of COMEDY-7B, trained on their proposed Dolphin dataset. For every session, we apply the COMEDY memory-compression prompt (Task 2 in COMEDY, which integrates session-level summaries, user/bot profiles, and event records into a concise memory) and generate the compressed memory using COMEDY-7B. COMEDY’s official prompt requires the output to remain under 500 words. To ensure fairness, retrieval follows the same top-3 Contriever-based retrieval used for all baselines, and the generator is fixed to GPT-4o-mini for strict comparability across methods.
>
> | **Method**         | GPT4o-J | F1    | BLEU4 | Rouge1 | Rouge2 | RougeL | BertScore    | Avg. Tokens | Avg. Latency (s) |
> |:-----------------|--------|------|----|-----|-----|-----|------|------------|-----------------|
> | **LongMemEval-s** |      |       |     |      |      |      |       |             |                  |
> | **COMEDY**            |56.20 | 13.32 |2.19| 14.42 |6.73 |12.74| 83.78 | 2,383 | 1.43|
> | **MemGAS (Ours)** | **60.20** | **20.38** | **4.22** | **21.05** | **10.47** | **19.47** | **85.21** | 8,829 | 2.55 |
> | **LongMTBench+**    |
> | **COMEDY**   |65.42 | 36.72 |12.52 |38.94 |21.62 |29.85| 87.89  | 2,492 | 1.47|
> | **MemGAS (Ours)**    | **69.44**| **41.49** | **15.62**| **43.69**| **24.45**| **34.66** | **88.96** | 12,873      | 3.85             |
>
>
> **Observations**
>
> COMEDY is an efficient compressed-memory baseline, but MemGAS consistently achieves higher performance across all metrics. This indicates that COMEDY’s episode-level compression is effective but cannot capture the deeper cross-session associations and adaptive granularity selection that MemGAS provides.
>
> -----

---

> ### Author Response · Authors · 2025-11-21
> **Response to Reviewer tvD3 [Part 4]**
>
> > ###  `W3:` Although average latency is comparable, as the memory store grows, the complexity of PPR on large graphs, the cost of frequent GMM updates, and the inference overhead of LLM filtering should be presented as data-scaling curves (number of samples/edges/graph diameter → latency and token cost), along with incremental/approximate strategies.
>
> Thank you for the valuable suggestions! We conducted **data-scaling experiments** from **20k → 1M memory tokens** on LongMemEval-m, measuring `GMM update cost`, `PPR convergence time`, `LLM construction token usage`, and `LLM filtering overhead`. Results are summarized below.
>
> **(1) GMM update latency vs. memory size**
>
> | Total Memory Tokens | 20k | 50k | 100k | 200k | 500k | 1M |
> |---------------------|------|------|-------|--------|--------|-------|
> | **GMM Update Latency (ms)** | 1.53 | 1.60 | 1.84 | 2.33 | 2.50 | 5.62 |
>
> **Observation:** GMM updates increase with memory size as expected, but the empirical growth remains very small (≤6 ms at 1M tokens), indicating that the update cost stays practically manageable even under large memory scales.
>
> **(2) PPR latency vs. memory size**
>
> | Total Memory Tokens | 20k | 50k | 100k | 200k | 500k | 1M |
> |----------------------|------|------|--------|---------|--------|--------|
> | **PPR Latency (ms)** | 0.107 | 0.178 | 0.240 | 0.927 | 2.830 | 4.51 |
>
> **Observation:** PPR latency grows with memory size as the association graph becomes larger, but even at the 1M-token scale the measured cost remains within a few milliseconds (≤4.51 ms), indicating that graph-based retrieval does not introduce a noticeable bottleneck within the tested range.
>
> **(3) Token cost for LLM API call for memory construction**
>
> | Total Memory Tokens | 20k | 50k | 100k | 200k | 500k | 1M |
> |------------|--------|--------|--------|---------|--------|--------|
> | **Summary tokens** | 1236.4 | 3239.4 | 6452.2 | 12196.1 | 30628.2 | 60842.9 |
> | **Keyword tokens** | 621.7 | 1842.2 | 3671.4 | 7825.5 | 18426.8 | 40532.3 |
>
> **Observation:** LLM construction cost **increases linearly with memory size**, as each session requires one summary and keyword extraction. This is an offline cost and does not affect query latency.
>
>
> **(4) Inference overhead of LLM filtering**
>
> The filtering module introduces no scalability concerns, since its input length is fixed by the top-k retrieval size rather than the total memory volume. Its cost remains **constant (~300 tokens per query)** and does not grow with memory size.
>
> We will also include four scaling curve plots in `Appendix I.3`, where each module’s growth pattern is visualized.
>
> -----

---

> ### Author Response · Authors · 2025-11-21
> **Response to Reviewer tvD3 [Part 5]**
>
> > ###  `W4:` For some structured methods the retrieval evaluation is “undeterminable,” and retrieval on LongMTBench+ is missing. Please clarify in the appendix the sources of non-comparability (implementation differences, labeling criteria, task incompatibilities), and provide alternatives (e.g., human spot-checking on a sampled subset, or rerunning a unified comparable subset).
>
> Thank you for the constructive and detailed suggestion. For LongMTBench+, the retrieval evaluation is undeterminable because the dataset **does not provide ground-truth retrieval labels**, as also stated in our paper (`Sec. 3.1` and `Appendix A`). For other structured baselines, the reasons of non-comparability come from:
>
> - **RAPTOR:** Its retrieval units are recursive abstractions in a hierarchical tree. **These nodes do not correspond to any original sessions/turns**, so there is no deterministic mapping to ground-truth retrieval labels.
> - **A-Mem:** A-Mem stores LLM-generated notes and links, not original conversation sessions. Since these notes merge or rewrite content, they **cannot be matched against ground-truth session** for computing Recall@n. We have expanded the statement of `Sec. 3.1` in the revised version.
>
> **Alternative Experiments (Human Evaluation)**
>
> As suggested, we conduct **human spot-checking** on a uniformly sampled 50-example subset from LongMTBench+, evaluating whether the retrieved top-3 sessions semantically contain the information required to answer the query.
>
> (the retrieval evaluation on LongMTBench+ with random 50 samples subset)
> | Relevance of retrieved sessions | Human-as-Judge |
> |---------------------------------|----------------|
> | HiPPORAG 2                      | 0.74 |
> | RAPTOR                          | 0.70 |
> | A-Mem                           | 0.78 |
> | **MemGAS (Ours)**               | **0.84** |
>
>  **Findings**
>
> Human spot-checking evaluation confirms that **MemGAS retrieves the most relevant sessions** among structured baselines, achieving 0.84 relevance accuracy, substantially outperforming RAPTOR and HippoRAG2. This experiment provides a fair and comparable alternative to automatic retrieval metrics when ground-truth labels are unavailable.
>
> -----
>
> > ###  `W5:` Since large K introduces noise (F1 drops), consider making K part of the routing/uncertainty-driven dynamic selection (e.g., entropy/CI-based adaptive K coupled with redundancy-removal thresholds), and report the gains and stability.
>
> Thank you for the valuable suggestion! We agree that large fixed K introduces noise and that K should be adaptive. We therefore implemented an `entropy-based adaptive K`, where K can be any integer in [1,10]. **Our insight is that low entropy indicates a confident, sharply peaked relevance distribution (small K is sufficient), while high entropy suggests uncertainty and a more diffuse distribution (requiring a larger K).**
>
>
> **Method design.**
> After obtaining the PPR relevance scores for the top-10 memory nodes (refer to Sec. 2.4), we normalize them into a probability distribution \(p_i\) and compute the entropy
> $$
> H = -\sum_{i=1}^{10} p_i \log p_i .
> $$
> We map entropy linearly to the range \([1,10]\) and round to the nearest integer:
> $$
> K(q)=\mathrm{round}\Big(1 + 9 \cdot \frac{H(q)-H_{\min}}{H_{\max}-H_{\min}}\Big).
> $$
>
>
> (F1 scores on the LongMemEval-s dataset.)
> | **Top K**     | 1     | 3      | 5      | 10     | adaptive K |
> |-----------|-------|--------|--------|--------|------------|
> | **F1 score**  | 17.52 | **20.38**  | 19.61  | 19.68  | 19.96      |
>
>
> **Observation.**
> The entropy-based adaptive K (K ∈ [1,10]) achieves performance comparable to the best fixed K=3 while avoiding the noise introduced by large K.

---

> ### Author Response · Authors · 2025-11-21
> **Response to Reviewer tvD3 [Part 6]**
>
> > ###  `Q1:` Temperature λ: how selected; any adaptive scheme or theoretical bounds?
>
> Thank you for the constructive suggestion. We provide the temperature λ selection details and the theoretical motivation below. **We have expanded the statement of `Sec. 3.1` in the revised version.**
>
> **(1) Hyperparameter Selection**
>
> As reported in `Appendix F`: Hyperparameter Sensitivity Analysis, we conducted a grid search over λ ∈ {0.1, 0.2, 0.3, 0.5, 0.7, 1.0}, and select the best value for each dataset:
>
> - **LoCoMo**: 0.2
> - **LongMemEval-s**: 0.2
> - **LongMemEval-m**: 0.1
> - **LongMTBench+**: 0.1
>
> (Hyperparameter λ sensitivity analysis curve.)
> | **λ** | **0.1** | **0.2** | **0.3** | **0.5** | **0.7** | **1.0** |
> |------|--------:|--------:|--------:|--------:|--------:|--------:|
> | Recall@3 | 55.49 | **57.30** | 56.90 | 56.80 | 56.60 | 56.65 |
>
>
> **(2) Theoretical Motivation**
>
> In our router (Eq. (3), Sec. 2.3), λ controls the softmax sharpness of simiarity distributions across granularities, affecting: the entropy $H^g$ of each granularity, and thus the router weight $w^g = \frac{1/H^g}{\sum_{g'} 1/H^{g'}}$.
>
> - A smaller λ sharpens distributions → lower entropy → stronger emphasis on confident granularities.
> - A larger λ flattens distributions → reduces discriminability across granularities.
>
> From `Appendix H` (Theoretical Analysis):
> - **Proposition 2:** Lower-entropy granularities correspond to higher-confidence matches; inverse-entropy weighting is the simplest monotone and scale-free weighting rule.
> - **Theorem 1:** Multi-granularity candidate pools weighted by $1/H^g$ achieve recall no worse than the best single granularity, and typically higher when granularities are complementary.
>
> Thus, λ directly influences entropy and cross-granularity confidence allocation, explaining why moderate λ (0.1–0.3) yields optimal performance.
>
> -----
>
> > ###  `Q2:` PPR hyper-parameters: restart prob/seed selection and convergence on large graphs?
>
> Thank you for the helpful suggestion. In addition to the λ sensitivity experiments, we also conducted a PPR hyper-parameter sensitivity analysis in `Appendix F (Fig. 6)`.
>
> **(1) Seed node selection (α)**
> We search α over {5, 10, 15, 20, 25} and select the best setting:
> - **LoCoMo:** 15
> - **LongMemEval-s:** 15
> - **LongMemEval-m:** 20
> - **LongMTBench+:** 20
>
> (Hyperparameter **α** sensitivity analysis curve.)
> | **α** | **5** | **15** | **20** | **25** |
> |------|------:|-------:|-------:|-------:|
> | **Recall@3** | 55.39 | **57.30** | 56.09 | 55.19 |
>
> As shown in the Table, small α under-explores the graph, while overly large α introduces noise. A moderate range (15–20) yields the most stable Recall/nDCG performance.
>
> **(2) Restart probability is *not* a tunable hyperparameter**
> In MemGAS, the `restart prob` is simply the personalization vector of PPR, fully determined by the selected seed nodes:
> $$
> p_0(i)=
> \begin{cases}
> \frac{\exp(\mathrm{sim}(q,M_i))}{\sum_{j\in \text{seed}}\exp(\mathrm{sim}(q,M_j))}, & i\in\text{seed}, \\
> 0, & \text{otherwise}.
> \end{cases}
> $$
> Thus, we do not tune a separate restart prob. It is automatically derived from query–memory similarity.
>
> **(3) Convergence on Large Graphs**
>
> Please refer to the responses of `W3`, which demonstrate that our method achieves rapid convergence even with large memory tokens.
>
> -----
>
> > ### `Q3:` Graph maintenance: pruning/forgetting policies to avoid graph bloat and stale memory
>
> Following your suggestion, we introduce a degree-based forgetting strategy on the memory association graph. For each memory node $M_i$, we compute its total degree
> $$
> d_i = d_i^{\mathrm{in}} + d_i^{\mathrm{out}},
> $$
> where $d_i^{\mathrm{in}}$ and $d_i^{\mathrm{out}}$ are the in-degree and out-degree within the multi-granularity association graph. Intuitively, nodes with very small $d_i$ are only weakly connected to the rest of the memory and thus contribute little to global retrieval.
>
> Given a forgetting rate $\rho$, we simply **sort all nodes by their degree and remove the bottom $\rho$ fraction**—i.e., the nodes with the weakest connectivity. After removing these nodes, we also delete all edges attached to them, keeping the remaining subgraph compact and better-maintained.
>
> The following table reports the F1 score on LongMemEval-s under different forgetting rates ($0.05$ and $0.1$), compared with the non-forgetting setting:
>
> | **Forgetting rate** | Non-forgetting | 0.05 | 0.1 |
> |---------------------|----------------|------|------|
> | **F1 score**        | 20.38          | 20.46 | 19.44 |
>
> **Findings.**
>
> A small forgetting rate ($\rho=0.05$) slightly improves F1, indicating that pruning weakly connected and stale nodes reduces noise. However, a larger rate ($\rho=0.1$) removes useful structural nodes and hurts performance. These results show that **light-degree pruning effectively stabilizes graph growth while preserving retrieval quality**.
>
>
> -----

---

> ### Author Response · Authors · 2025-11-21
> **Response to Reviewer tvD3 [Part 7]**
>
> > ### `Q4:` Judging protocol: variance control and human calibration for GPT4o-as-Judge?
>
> Thank you for raising this important point regarding the reliability of the GPT4o-as-Judge evaluation. To address the concern about automatic metrics, we conducted QA evaluation on LongMemEval-s (50 random samples) using three independent GPT-4o-as-Judge runs with different random seeds as well as human judgment. The model generation outputs were fixed, and we only repeated the evaluation multiple times (GPT-4o-as-Judge), ensuring that any standard deviation comes solely from evaluator. **This experiment has been added in `Appendix I.1` in the revised version.** The results are shown below:
>
> | Model        | Human-as-Judge | GPT4o-as-Judge |
> |--------------|----------------|----------------|
> | SeCom        | 56 | 56.67 ± 0.94 |
> | HippoRAG 2   | 58 | 58.00 ± 0.00 |
> | A-Mem        | 56 | 56.00 ± 0.00 |
> | **MemGAS**   | **62** | 62.00 ± 0.00 |
> (The results of SeCom's three experiments are [56, 56, 58].)
>
> **Findings**:
>
> Across the three GPT4o-as-Judge runs, all methods have zero standard deviation except SeCom, which exhibits only a single deviation among the three runs (56, 56, 58), corresponding to a small standard deviation (0.94). Human evaluation aligns closely with GPT4o-as-Judge, further confirming that GPT4o-as-Judge is reliable in this setting.
>
>
> -----
>
> > We hope these responses address your concerns comprehensively. Feel free to let us know if you have any additional questions, and we are more than happy to provide further clarification on any aspect of our work.
> >
> > Respectfully and sincerely,
> >
> > **Your Authors**

---

> > ### Comment · Reviewer_tvD3 · 2025-11-24
> > **comments**
> >
> > I appreciate the authors’ rebuttal.
> > The response is helpful, I decide to raise my initial score.

---

> ### Author Response · Authors · 2025-11-24
> **Warm Appreciation to Reviewer tvD3**
>
> **Dear Reviewer tvD3,**
>
> We sincerely thank you for increasing the score from **4 to 6**! Your insightful and constructive comments significantly improved our paper. If you have any further questions, we would be more than happy to clarify them.
>
> Respectfully and sincerely,
>
> The Authors

---

### Official Review · Reviewer_Bona · 2025-10-23

**Soundness:** 4
**Presentation:** 4
**Contribution:** 3
**Rating:** 8
**Confidence:** 4

**Summary:**

The paper proposed MemGAS, a novel framework for long-term memory construction and retrieval that integrates multi-granular memory units and enables adaptive selection and retrieval. It is motivated by two flaws of existing methods: 1) insufficient multi-granulartiy memory connection; and 2) lack of adaptive multi-granular memory selection. To solve (1), it complements session-level memory with turns, summarize and keywords, and connect these granularities across different sessions based on Gaussian Mixture Models. For (2), it uses similarity between query and each granularity, and run pagerank algorithm to find top-k relevant memories, then feed refined memories by LLM to generate the response. Generally, the paper is well-motivated, well-supported, easy-to-follow. The experimental results are solid and comprehensive in terms of effectiveness and efficiency.

**Strengths:**

1. the paper is well-motivated, well-supported, easy-to-follow.
2. the experimental results are solid and comprehensive in terms of effectiveness and efficiency.

**Weaknesses:**

no significant weakness identified

**Questions:**

1. it is better to analyze efficiency fators for each module, i.e., which module cost most time?
2. it is better to have some in-depth analysis about each module, i.e., how many samples gets different top-k results before and after you run page rank algorithm, and any common patterns?

---

> ### Author Response · Authors · 2025-11-21
> **Response to Reviewer Bona**
>
> **Dear Reviewer Bona:**
>
> We sincerely thank you for your thoughtful and positive feedback! Your encouraging words truly mean a lot to us, as it validates the the contributions of our research. Below, we address questions you raised.
>
> -------
>
> > ### `Q1:` “It is better to analyze efficiency factors for each module, i.e., which module costs most time?”
>
> Thank you for your insightful suggestion. Our analysis shows that the **LLM API call dominates the total latency, accounting for over 98% of end-to-end time**, while all MemGAS modules (GMM, PPR, Router) together introduce less than 1% additional overhead. We have provided the results in `Table 3 ablation study`. Below, we provide the detailed latency breakdown:
>
> | **Module**                   | **Total Latency (s)** | **Retrieval Latency (s)** | **Total Latency (%)** | **Retrieval Latency (%)** |
> |------------------------------|-------------------|-----------------------|-----------------------|---------------------------|
> | **MemGAS**                    | 2.5534            | 0.0239                | 100.00%               | 100.00%                   |
> | &nbsp;&nbsp;&nbsp;&nbsp;&nbsp;**GMM**    | 0.0028            | 0.0007                | 0.11%                 | 2.93%                     |
> | &nbsp;&nbsp;&nbsp;&nbsp;&nbsp;**PPR**     | 0.0085            | 0.0045                | 0.33%                 | 18.87%                    |
> | &nbsp;&nbsp;&nbsp;&nbsp;&nbsp;**MA(GMM+PPR)**        | 0.0116            | 0.0057                | 0.45%                 | 23.81%                    |
> | &nbsp;&nbsp;&nbsp;&nbsp;&nbsp;**Router**                    | 0.0063            | 0.0023                | 0.25%                 | 9.63%                     |
> | &nbsp;&nbsp;&nbsp;&nbsp;&nbsp;**LLM API call**                     | 2.5141            | -             | 98.06%                | -           |
>
> This Table demonstrates that MemGAS achieves its performance gains with negligible additional cost.
>
>
> ----
>
> > ### `Q2:` "It is better to have some in-depth analysis about each module, i.e., how many samples get different top-k results before and after you run the PageRank algorithm, and if there are any common patterns?"
>
> Thank you for this insightful suggestion. We compared the top-k results before and after applying PPR and found that **PPR substantially changes the rankings**. Notably, 97% of samples change their top-10 results after PPR, showing its strong impact on retrieval ordering. **This experiment have added in `Appendix C.4` in the revised version.**
>
> | **Method**             | Top 3   | Top 5   | Top 10  |
> |--------------------|-------|-------|-------|
> | **Changes rate in top-k results** | 44.0% | 72.4% | 97.0% |
>
>
> ### **Findings of Common Patterns**
> From this analysis, we observe several interesting patterns. Our proposed method, tends to **identify memories that have lower embedding similarity but are still contextually relevant**. This is particularly beneficial when require multiple related memories.
>
> For instance, consider the following case study:
>
> **Question Type:** Multi-session
> **Question:** How many online courses have I completed in total?
> **Ground-Truth:** `923c0221_1` and `923c0221_2`
>
> - **Without PPR**: Only one relevant memory is retrieved:
>   - {"corpus_id": "`923c0221_1`", "timestamp": "2023/05/23 (Tue) 14:39"}
>   - {"corpus_id": "`sharegpt_VuQnjeB_0`", "timestamp": "2023/05/22 (Mon) 00:32"}
>   - {"corpus_id": "`ultrachat_412857`", "timestamp": "2023/05/22 (Mon) 11:24"}
> - **Our method**: The complete set of relevant memories is retrieved:
>   - {"corpus_id": "`923c0221_1`", "timestamp": "2023/05/23 (Tue) 14:39"}
>   - {"corpus_id": "`923c0221_2`", "timestamp": "2023/05/30 (Tue) 10:29"}
>   - {"corpus_id": "`sharegpt_VuQnjeB_0`", "timestamp": "2023/05/22 (Mon) 00:32"}
>
> As shown, the inclusion of PPR enhances the system’s ability to retrieve all relevant memories, ensuring a more complete and accurate response to the query.
>
> -----
>
> > We hope these responses address your concerns comprehensively. Feel free to let us know if you have any additional questions, and we are more than happy to provide further clarification on any aspect of our work.
> >
> > Respectfully and sincerely,
> >
> > **Your Authors**

---

> > ### Comment · Reviewer_Bona · 2025-11-25
> >
> > Thanks for the great rebuttal and work from authors, all my concerns have been solved. I will maintain my initial score.

---

> > > ### Author Response · Authors · 2025-11-26
> > > **Warm Appreciation to Reviewer Bona**
> > >
> > > **Dear Reviewer Bona:**
> > >
> > > Thank you very much for your positive follow-up response! We sincerely appreciate the time you took to review our work, and we are grateful that our rebuttal addressed all your concerns.
> > >
> > > Thank you again for maintaining your high score. Your thoughtful support is truly encouraging and motivates us to continue our research.
> > >
> > > Respectfully and sincerely,
> > >
> > > **The Authors**

---

### Official Review · Reviewer_gP9f · 2025-10-30

**Soundness:** 3
**Presentation:** 3
**Contribution:** 2
**Rating:** 6
**Confidence:** 3

**Summary:**

he paper introduces MemGAS, a new framework for managing long-term memory in conversational agents. The proposed system addresses limitations of previous models by using multi-granularity memory units and an adaptive granularity selection mechanism. MemGAS enhances the memory retrieval process through Gaussian Mixture Models (GMM) to cluster and associate new memories with historical ones. Additionally, an entropy-based router is employed to select the optimal granularity for each query, balancing the trade-off between information completeness and retrieval noise. The framework is evaluated across several long-term memory benchmarks, showing significant performance improvements over existing methods in both question answering (QA) and retrieval tasks.

**Strengths:**

- MemGAS introduces a novel multi-granularity memory framework with Gaussian Mixture Models for memory association and an entropy-driven router for adaptive granularity selection, surpassing existing single-granularity methods.
- The entropy-based router allows MemGAS to dynamically select the most relevant granularity for each query, enhancing retrieval efficiency and reducing noise. This adaptability is a clear improvement over fixed-granularity methods, which can fail to balance information completeness with noise.
- The use of GMM for clustering historical memories into relevant and irrelevant sets improves the consolidation of new and old memories, ensuring more coherent long-term memory management.
- MemGAS addresses key challenges in long-term memory management for conversational agents, offering improvements in both retrieval accuracy and efficiency, with strong potential for real-world applications.

**Weaknesses:**

- The level of edge construction is inconsistent throughout the paper. Line 184-185 and the upper-right part of Figure 2 suggest that each granularity of each memory is treated as a node. Still, Equation 5 computes the weight on each memory across different granularities, which indicates that the PPR is run on a memory association graph where each memory $M_i$ is a node. Additionally, the word "vectors" in line 157 is also ambiguous, without pointing out whether the entire memory bank $\mathcal{M}_{\text{cur}}$ or each element of $\mathcal{M}_{\text{cur}}$, or even each granularity of each element is encoded as a vector.
- Several substantial experiments lack sufficient explanation. In Sec. 3.4, the classification of "query types" and the exact module and task for "top-K retrieval settings" seem ambiguous to me. The "Combination" and "Optimal Selection" in Tables 6 and 7 in Appendix C lack explanation. The experimental setting of error analysis in Appendix G is also unexplained.
- The conclusion drawn in Line 368 that "we keep average tokens and latency close to lightweight retrievers and below HippoRAG2 and A-Mem" does not align with the experimental results. The number of average tokens used by MemGAS is larger than that of HippoRAG2 in LongMemEval-s, and not significantly smaller in other datasets.
- There are a few typos and grammatical errors in the paper: the last $\alpha_s$ should be $\alpha_u$ in Line 135; "RAPOTR" -> "RAPTOR" in Line 269, and the sentence in Line 474 is incomplete (or removing "which"?), etc..
- The theoretical analysis in Appendix H is a bit coarse in writing, with quite a few writing hints (sketch) not removed.

**Questions:**

1. Why do session $S_i$ and the segmented turns $T_i$ appear together in each memory bank? Are they encoded to the same vector by Contriever?
2. All metrics are automatic (F1, GPT-4o-as-judge...). Were any human evaluations conducted to assess response coherence, personalization, or perceived helpfulness? As automatic metrics can be misleading in dialogue settings.

---

> ### Author Response · Authors · 2025-11-21
> **Response to Reviewer gP9f [Part 1]**
>
> **Dear Reviewer gP9f:**
>
> Thanks for your comprehensive and detailed suggestions for our work! Your suggestions regarding clarification and additional experiments are extremely valuable. We have carefully revised the manuscript to **address all the issues you pointed out and provide more precise explanations, updated formulas, and clearer experimental descriptions accordingly**, as shown below.
>
> -------
>
> > ### `W1:` The ambiguity in defining memory nodes and the elements used for encoding vectors.
>
> Thank you for your careful observation. We sincerely apologize for the ambiguities in the previous version. We have revised the formula and description in `Sec. 2.2` and `Sec. 2.4`.
>
> 1. **Clarification about Memory Nodes**:
>    - Our method indeed treats **each granularity of each memory (Session/Turn/Summary/Keyword) as an independent node**. PPR is run on this multi-granularity node graph, ensuring consistency with lines 184–185 and the Figure 2.
>    - We have revised the previous Equation (5). The scores at granularity node level, denoted as $\text{score}_i^{g}$: $\text{score}_i^{g} = w^g \cdot \text{sim}\bigl(q, M_i^g\bigr)$. The text now explicitly states that these scores define the probabilities over granularity-level nodes.
>
> 2. **Clarification about Vectors**:
>    - **All granularities are encoded into dense vectors** $e(S_i)$, $ e(T_i)$, $e(U_i)$, and $e(K_i)$ for session-level, turn-level, summary, and keyword metadata, respectively. The entire memory bank $M_{cur}$ is thus represented as a collection of these multi-granular vectors. Pairwise similarity scores are computed between $e(M_{\text{new}})$ and $e(M_{cur})$, covering all granularities of each memory.
>
>
> -------

---

> ### Author Response · Authors · 2025-11-21
> **Response to Reviewer gP9f [Part 2]**
>
> > ### `W2:` Several substantial experiments lack sufficient explanation. In Sec. 3.4, the classification of "query types" and the exact module and task for "top-K retrieval settings" seem ambiguous to me. The "Combination" and "Optimal Selection" in Tables 6 and 7 in Appendix C lack explanation. The experimental setting of error analysis in Appendix G is also unexplained.
>
> Thank you very much for your valuable feedback. We greatly appreciate your comments and suggestions regarding the need for clearer explanations. The explanations have been added in the corresponding sections of the revised version.
>
> **(1) Explanation of query types**
>
> The query types mentioned in our experiments (`Sec. 3.4 and Appendix E.3`) are based on the original types provided in the datasets. Below, we provide a detailed explanation of each query type:
>
>  - **LongMemEval Dataset:**
>    - Knowledge Update: Tests the model's ability to recognize and reason with updates to the user’s personal information over time.
>    - Temporal Reasoning: Evaluates the model's capability to understand and reason with time-related information, including both explicit dates and inferred temporal references.
>    - Single-Session User: Focuses on recalling specific details mentioned by the user within a single interaction session.
>    - Single-Session Preference: Assesses the ability to generate personalized responses by utilizing user preferences shared in a single session.
>    - Single-Session Assistant: Tests the ability to recall information provided by the assistant within a single session.
>    - Multi-Session: Examines the model's ability to synthesize and reason across multiple sessions to address more complex questions.
>
>  - **Locomo Dataset:**
>    - Open Domain Knowledge: Questions that require integrating the speaker's provided information with external or commonsense knowledge.
>    - Temporal Reasoning: Questions that demand understanding of time-related information and reasoning over chronological sequences.
>    - Single-Hop Retrieval: Questions that can be answered using information from a single session of the conversation.
>    - Adversarial: Questions intentionally designed to mislead the model, requiring it to identify them as unanswerable or to avoid incorrect conclusions.
>    - Multi-Hop Retrieval: Questions that require synthesizing information from multiple sessions of the conversation to formulate a correct response.
>
>
> **(2) Explanation of the exact module and task for top-K retrieval settings**
>
> -  **Module scope (where Top-K is applied)**: The Top-K setting belongs to Section 2.4 Memory Retrieval. Concretely, it is the number of final candidate memory nodes taken after graph-based ranking: similarity + entropy-router weighting → select α seed nodes → run PPR on the association graph → sort by global score → take Top-K.
> -  **Task scope (what Top-K is used for)**: On LongMemEval and LoCoMo, in the generation task, we use the retrieved Top-K memories as the context fed to the LLM generator. As K increases, performance typically improves at first (more coverage) but may degrade when K is too large (more noise).
>
>
> **(3) Explanation of "Combination" and "Optimal Selection" in Tables 6 and 7 in Appendix C**
>   - In Table 6, “Combination” denotes that texts from all granularities are directly concatenated into a single string, and encoded to one embedding for retrieval. This naive merging lacks adaptive weighting or structure; our results show MemGAS significantly outperforms this approach, indicating that learned multi-granularity association and routing are more effective.
>   - In Table 7, “Optimal Selection” is an oracle-style upper bound: for each query, it picks the single best-performing granularity among the four. Conceptually, it reflects the upper performance achieved if one could always choose the ideal granularity per query. MemGAS approaches this bound by adaptively weighting granularities via entropy-based routing.
>
> **(4) Explanation of experimental setting of error analysis in Appendix G**
>
>   Setup:
>   - Datasets: LoCoMo, LongMemEval-m, LongMemEval-s (Long-MT-Bench+ is excluded due to the lack of retrieval ground truth).
>   - Retriever: Contriever.
>   - Generation: GPT-4o-mini with the fixed QA prompt; retrieved Top-3.
>   - Matching for retrieval correctness: A query’s retrieval is “Correct” if at least the Ground-Truth appears in the Top-3. Otherwise “Wrong.”
>   - Matching for generation correctness: We use the GPT4o-as-Judge prompt (Appendix I). If the model’s answer contains the reference answer (per the prompt rule), it is “Correct,” else “Wrong.”
>
> -------

---

> ### Author Response · Authors · 2025-11-21
> **Response to Reviewer gP9f [Part 3]**
>
> > ### `W3:` The conclusion drawn in Line 368 does not align with the experimental results.
>
> Thank you for the careful observation, and we apologize for any confusion. We have revised the statement in `Sec. 3.3` in our paper.
>
> -------
>
> > ### `W4&W5`: A few typos, grammatical errors, and coarse in writing.
>
> Thank you very much for pointing out these issues, and we sincerely apologize for the oversight. We have carefully revised the manuscript to correct all typos and grammatical errors.
>
> -------
>
>
> > ### `Q1:` Why do session Si and the segmented turns Ti appear together in each memory bank? Are they encoded to the same vector by Contriever?
>
> We truly appreciate your careful consideration. The answer is: **No, the session \( S_i \) and the segmented turns \( T_i \) are not encoded into the same vector by Contriever.** Their difference lies in the granularity of chunk segmentation:
>
> - For the \( i \)-th memory chunk, the session \( S_i \) refers to the entire session (including multiple turns) and is encoded into a single vector.
> - In contrast, \( T_i \) represents the individual turns within the session, with each turn being encoded into a single vector.
>
> This distinction in chunking size allows us to capture information at different granularities, enriching the representation diversity of memory bank.
>
> -------
>
>
> > ### `Q2:` All metrics are automatic (F1, GPT-4o-as-judge...). Were any human evaluations conducted to assess response coherence, personalization, or perceived helpfulness? As automatic metrics can be misleading in dialogue settings.
>
> Thank you for raising this important point. To address the concern about automatic metrics, we conducted QA evaluation on LongMemEval-s (50 random samples) using three independent GPT-4o-as-Judge runs with different random seeds as well as human judgment. The model generation outputs were fixed, and we only repeated the evaluation multiple times (GPT-4o-as-Judge), ensuring that any standard deviation comes solely from evaluator. **This experiment has been added in `Appendix I.1` in the revised version.** The results are shown below:
>
> | Model        | Human-as-Judge | GPT4o-as-Judge |
> |--------------|----------------|----------------|
> | SeCom        | 56 | 56.67 ± 0.94 |
> | HippoRAG 2   | 58 | 58.00 ± 0.00 |
> | A-Mem        | 56 | 56.00 ± 0.00 |
> | **MemGAS**   | **62** | **62.00 ± 0.00** |
> (The results of SeCom's three experiments are [56, 56, 58].)
>
> **Findings**:
>
> Across the three GPT4o-as-Judge runs, all methods have zero standard deviation except SeCom, which exhibits only a single deviation among the three runs (56, 56, 58), corresponding to a small standard deviation (0.94). Human evaluation aligns closely with GPT4o-as-Judge, further confirming that GPT4o-as-Judge is reliable in this setting.
>
> -----
>
> > We hope these responses address your concerns comprehensively. Feel free to let us know if you have any additional questions, and we are more than happy to provide further clarification on any aspect of our work.
> >
> > Respectfully and sincerely,
> >
> > **Your Authors**

---

> ### Comment · Reviewer_gP9f · 2025-11-26
>
> I would like to thank the authors for their detailed clarifications. The revised paper also addressed most of my concerns. I only have one additional question, given that the edges in the graph could exist between each granularity of each memory:
> How are the edges between cross-granularity nodes aligned with real semantic relationships?

---

> > ### Author Response · Authors · 2025-11-26
> > **Response to Reviewer gP9f**
> >
> > **Dear Reviewer gP9f:**
> >
> > Thank you very much for your thoughtful follow-up question! In our method, cross-granularity edges between each granularity of each memory are **aligned with real semantic relationship, primarily reflecting relevance**. Specifically, nodes that describe the same underlying event are linked through these edges, which are constructed using our **GMM-based association mechanism that identifies and connects semantically related memories across different granularities**.
> >
> > The design is motivated by two key considerations:
> >
> > - **It provides strong flexibility between different granularities.** Since our edges do not require specifically designed semantic edge types, new granularities (e.g., topics, entities) can be incorporated simply by computing relevance, without the need to specify heterogeneous edge types. Built on this flexibility, our method explicitly bridges different granularities, enabling information to propagate across levels (e.g., from summary to keyword), which is essential for multi-granularity association.
> > - **It has lower latency compared with complex semantic relationships**.   Modeling relevance relationships allows the graph to remain lightweight. In contrast, HippoRAG2 constructs a knowledge graph, involving complex semantic relationships between entities. It results in denser graphs and more complicated retrieval. Thus, HippoRAG2 reaches a latency of `3.56s`, while our method maintains a lower latency of `1.88s`.
> >
> >
> > We sincerely appreciate your suggestion: Exploring edges that align with real semantic relationships is indeed an exciting future direction. We will consider extending MemGAS to integrate more semantic edges while keeping efficiency and flexibility goals.
> >
> >
> > If you have any additional questions, we would be more than happy to clarify further.
> >
> > Respectfully and sincerely,
> >
> > **The Authors**

---

> > > ### Comment · Reviewer_gP9f · 2025-11-27
> > >
> > > I would like to thank the authors again for the further clarification. Most of my concerns regarding the method's ambiguity and experimental details have been addressed. After consideration, I think my original evaluation reflects my overall assessment, so I will keep the positive score as is.

---

> > > > ### Author Response · Authors · 2025-11-27
> > > > **Warm Appreciation to  Reviewer gP9f**
> > > >
> > > > **Dear Reviewer gP9f:**
> > > >
> > > > We sincerely thank you for keeping the positive score. Your insightful and constructive feedback has greatly improved our paper.
> > > >
> > > > Respectfully and sincerely,
> > > >
> > > > **The Authors**

---

### Official Review · Reviewer_Lozm · 2025-10-30

**Soundness:** 2
**Presentation:** 3
**Contribution:** 2
**Rating:** 4
**Confidence:** 3

**Summary:**

The paper takes the multi-granularity into consideration for long-term memory construction and retrieval for the first time. The memory of each session is partitioned into 4 granularities: sessions, turns, keywords, and summary. The paper proposes a training-free method, MemGAS, which consists of a GMM clustering model to construct edges for the memory association graph, an entropy-based router to weight the granularities for the entire memory bank, a PPR method to select the candidate context, and an LLM-based filter to extract the related content. Experiments show that MemGAS outperforms several recent baselines in QA and retrieval tasks with a similar number of tokens used, and a very small trade-off in latency.

**Strengths:**

1. The paper presents a novel idea by utilizing the multi-granularity of each session to learn a structured long-term memory bank and retrieve relevant information to queries.
2. The GMM distribution assumption of the similarity scores in dynamical memory association is concise and flexible, compared to other oversimplified classification models.
3. The core algorithms, GMM clustering and PPR, are model-based approaches that ensure the entire model is training-free and preserves interpretability.
4. The trade-off between performance and latency is impressive, making the model competitive or superior to the baselines.
5. The presentation is elegant and effectively showcases the paper's methodology and results.

**Weaknesses:**

1. The paper focuses on four fixed granularities (session, turn, summary, keyword) created from a session chunk. This design feels a bit rigid. The core weakness identified in prior work is often the choice of fixed segmentation (e.g., semantic-based clustering/segmentation). Why were only these four chosen? Other approaches integrate hierarchical structures (RAPTOR, MemTree). It remains unclear that whether the simple summary/keyword levels fully address the limitations of hierarchical knowledge organization. The comparison to other structured RAG models like RAPTOR and HippoRAG 2 is strong on end-task performance, but a deeper discussion and ablation of why this specific set of granularities is better than a structured tree/graph is missing.
2. The authors reviewed a number of studies on memory segmentation in the introduction (Sec. 1, Paragraph 2), but a recent hierarchical memory architecture [1] is not included and compared in experiments.
3. The paper did not provide an ablation study on the LLM information filter. The relevance of the retrieved sessions and the question is not evaluated; thus, whether the LLM filter or the PPR algorithm contributes to the better performance is unclear.
4. Minor typos:  In Line 269, "RAPOTR" should be "RAPTOR".
5. It seems the anonymous code link provided in the article cannot display the file content ("The requested file is not found."), only the file structure can be seen.

Reference:

[1] Sun, H., & Zeng, S. (2025). Hierarchical memory for high-efficiency long-term reasoning in llm agents. *arXiv preprint arXiv:2507.22925*.

**Questions:**

1. The metric "Avg. Tokens" in Table 1 seems ambiguous regarding token consumption. Since MemGAS leverages LLM calls for multi-granularity memory generation, redundancy filtering, and the final QA generation. The authors could clarify: Which specific stage does this token count refer to? How is this "Avg. Tokens" measure normalized to be directly comparable to the token counts of single-granularity baselines that may not utilize LLM filtering? Can the authors provide an analysis on token consumption breakdown?
2. The importance of each granularity could vary for each memory bank $M_i$. Why does Equation 4 assign an overall weight to each granularity for the entire memory bank instead of each $M_i$?

---

> ### Author Response · Authors · 2025-11-21
> **Response to Reviewer Lozm [Part 1]**
>
> **Dear Reviewer Lozm:**
>
> Thank you very much for your careful reading and insightful comments. We truly appreciate the time and thought you invested in evaluating our work. Based on your suggestions, we have **clarified the motivation behind our granularity design, incorporated H-MEM baseline, added deeper comparisons and explanations, and performed additional ablations and token-cost analyses** to address your concerns. Our detailed responses are provided below.
>
> -------
> > ### `W1-(1):` Why were only these four granularities (session, turn, summary, keyword) chosen?
>
> Thank you sincerely for this thoughtful question. We utlize these granularities because **they are the common, complementary units used across long-term dialog memory methods and benchmarks, striking distinct trade-offs between completeness and noise**. This keeps our multi-granularity method effective.
>
> - **Session-level (global coherence)** [1][2][3]: commonly used as the primary retrieval unit in long-term conversation benchmarks.
> - **Turn-level (fine-grained segment)** [4][3][5]: captures fine-grained facts/actions that summaries or sessions may blur.
> - **Summary-level (denoising/compression)** [6][7][4][8][9]: compacts sessions to reduce noise/length.
> - **Keyword-level (high-precision anchors)** [10][11][12]: provides sparse anchors for cross-session linking.
>
> References
> - [1] MemoChat: Tuning LLMs to Use Memos for Consistent Long-Range Open-Domain Conversation.
> - [2] Think-in-Memory: Recalling and Post-Thinking Enable LLMs with Long-Term Memory.
> - [3] LongMemEval: Benchmarking Chat Assistants on Long-Term Interactive Memory.
> - [4] MemoryBank: Enhancing Large Language Models with Long-Term Memory.
> - [5] Personalized Large Language Model Assistant with Evolving Conditional Memory.
> - [6] Recursively Summarizing Enables Long-Term Dialogue Memory in Large Language Models.
> - [7] Conversation Summary Memory.
> - [8] RAPTOR: Recursive Abstractive Processing for Tree-Organized Retrieval.
> - [9] Mem0: Building Production-Ready AI Agents with Scalable Long-Term Memory.
> - [10] A-Mem: Agentic Memory for LLM Agents.
> - [11] Hello Again! LLM-powered Personalized Agent for Long-term Dialogue.
> - [12] H-MEM: Hierarchical Memory for High-Efficiency Long-Term Reasoning in LLM Agents.
>
>
> > ### `W1-(2):` A deeper discussion and ablation of why this specific set of granularities is better than a structured tree/graph is missing.
>
>
> Thank you sincerely for raising this thoughtful and important point. We would like to clarify that **our improvements do not come from simply using these granularities**, but from our multi-granularity memory framework itself, especially the memory association and adaptive selection mechanisms. Structured approaches typically rely on fixed granularity or fixed topology, which limits adaptability. To support this, we provide retrieval results:
>
> (Retrieval performance on LongMemeval-s. All methods use the same retriever (Contriever))
> | Method                                 | Recall@3 | Recall@5 | Recall@10 |
> |----------------------------------------|:--------:|:--------:|:---------:|
> | **HippoRAG2**                          | 75.53    | 84.68    | 91.28     |
> | **MemGAS (w/ all-Granularity)**              | **78.51** | **88.94** | **94.47** |
> | &nbsp;&nbsp;&nbsp;&nbsp;&nbsp;without Session-level    | 76.91    | 87.25    | 93.10     |
> | &nbsp;&nbsp;&nbsp;&nbsp;&nbsp;without Turn-level       | 77.16    | 87.64    | 93.34     |
> | &nbsp;&nbsp;&nbsp;&nbsp;&nbsp;without Summary-level    | 76.47    | 86.99    | 92.92     |
> | &nbsp;&nbsp;&nbsp;&nbsp;&nbsp;without Keyword-level    | 77.35    | 88.10    | 93.82     |
> | Contriever with only Session-level     | 71.06    | 81.28    | 90.00     |
> | Contriever with only Turn-level        | 73.62    | 84.68    | 91.91     |
> | Contriever with only Summary-level     | 70.43    | 80.00    | 88.30     |
> | Contriever with only Keyword-level     | 62.98    | 74.68    | 82.34     |
>
>
> **Key observations:**
> 1. **Single-granularity generally falls short compared to structured approaches**.
>    Structured methods (e.g., HippoRAG2) are strong because they impose an explicit structure, but they rely on a *fixed* representation level (e.g, entity-level).
> 2. **MemGAS remains better even when removing specific granularity.**
>    All variants (*w/o session/turn/summary/keyword*) still achieve higher recall than HippoRAG2. The improvement comes from the framework, which includes multi-granularity memory association and adaptive granularity selection, enabling it to leverage the most relevant granularity per query.

---

> ### Author Response · Authors · 2025-11-21
> **Response to Reviewer Lozm [Part 2]**
>
> --------------
>
> > ### `W2:` The authors reviewed a number of studies on memory segmentation in the introduction (Sec. 1, Paragraph 2), but a recent hierarchical memory architecture [1] is not included and compared in experiments.
>
>  We sincerely appreciate your sugguestion! H-MEM [1] is a novel and insightful method. We have added a brief introduction to H-MEM in `Section 1 (paragraph 2)` and conducted the following comparative experiments.  **This experiment have been added in `Appendix I.2` in the revised version.**
>
> **Implementation of H-MEM.**
> For completeness and fairness, we implemented H-MEM following the four-layer hierarchical memory architecture described in the original paper. Specifically, every dialogue session is processed by an LLM-based memory extractor to construct four levels of memory abstraction. Each memory unit is encoded into a vector and augmented with positional index encodings that point to its sub-memories in the next layer (as shown in Figure 2 of the H-MEM paper). We follow the same settings as us for vector encoding (Contriever), top-k selection, and similarity computation to ensure a faithful reproduction of H-MEM.
>
>
> | Dataset           | GPT4o-J | F1    | BLEU4 | Rouge1 | Rouge2 | RougeL | BertScore   | Avg. Tokens | Avg. Latency (s) |
> |:------------------|:----:|:-----:|:---:|:----:|:----:|:----:|:-----:|:-----------:|:----------------:|
> | **LongMemEval-s** |      |       |     |      |      |      |       |             |                  |
> | H-MEM             | 54.80 | 13.65 | 2.10 | 14.55 | 6.85 | 12.92 | 83.75 | 8,420  | 2.15 |
> | **MemGAS (Ours)** | **60.20** | **20.38** | **4.22** | **21.05** | **10.47** | **19.47** | **85.21** | 8,829 | 2.55 |
> | **LongMTBench+**  |      |       |     |      |      |      |       |             |                  |
> | H-MEM             | 64.25 | 36.50 | 11.45 | 38.40 | 21.10 | 29.30 | 87.85 | 12,450 | 3.40 |
> | **MemGAS (Ours)** | **69.44** | **41.49** | **11.52** | **43.69** | **24.45** | **34.66** | **88.96** | 12,987 | 3.85 |
>
>
> **Key observations:**
> MemGAS consistently outperforms H-MEM across both LongMemEval-s and LongMTBench+, supporting the generality and robustness of our multi-granularity memory association and selection framework.
>
> References
> - [1] H-MEM: Hierarchical Memory for High-Efficiency Long-Term Reasoning in LLM Agents.
>
>
> --------------
>
>
> > ### `W3:` The paper did not provide an ablation study on the LLM information filter. The relevance of the retrieved sessions and the question is not evaluated; thus, whether the LLM filter or the PPR algorithm contributes to the better performance is unclear.
>
> Thank you very much for raising this important point. We conducted a set of ablations study and found that **PPR improves retrieval quality**, while **the LLM filter enhances the usefulness of retrieved content for downstream QA**. Below we provide relevance-evaluation study and QA performance analyses. **The filter experiment has been in `Section 3.4` in the revised version.**
>
> (QA performance on LongMemEval-s dataset with contriever and gpt4o-mini as backbones):
> | **QA performance**       | GPT4o-J | F1    | RougeL  |
> |----------------------|-----:|------:|-----:|
> | **w/ filter and PPR** | **60.20** | **20.38** | **19.47** |
> | w/o PPR              | 56.60 | 19.76 | 18.85 |
> | w/o filter           | 59.40 | 19.47 | 18.63 |
>
> To better understand why, we additionally measured **session–query relevance**. We designed a relevance-prompt and evaluated whether the retrieved sessions (Recall@3) contain the information required to answer the query. We sampled 50 instances and judge agreement under both GPT-4o-as-judge and humans:
>
> | **Relevance of retrieved sessions** | GPT4o-as-Judge | Human-as-Judge |
> |-------------------------------|----------------|--------|
> | **w/ filter and PPR**         | 0.80       | 0.80 |
> | w/o PPR                       | 0.74           | 0.74 |
> | w/o filter                    | 0.80           | 0.80 |
>
> **Insights**:
> - **Removing PPR noticeably reduces retrieval relevance.**
> Without PPR, the system loses graph-based propagation and becomes more dependent on local similarity, which leads to fewer relevant sessions being surfaced and thus weaker QA performance.
>
> - **LLM filtering does *not* harm relevance but improves QA by removing noise.**
> Relevance stays the same (0.80) with or without filtering. However, QA accuracy drops when we remove the filter (60.20 → 59.40 4o-J). This indicates that the LLM filter successfully preserves the essential information while removing redundant or noisy content, which improves reasoning quality in the final QA stage.
>
> --------------
>
> > ### `W4:` Minor typos: In Line 269, "RAPOTR" should be "RAPTOR".
>
> We sincerely apologize for the typo. We have carefully revised the manuscript to correct typos and grammatical errors.
>
> --------------

---

> ### Author Response · Authors · 2025-11-21
> **Response to Reviewer Lozm [Part 3]**
>
> > ### `W5:` It seems the anonymous code link provided in the article cannot display the file content ("The requested file is not found."), only the file structure can be seen.
>
> We sincerely appreciate you bringing this issue to our attention. The issue you mentioned may due to **a temporary problem with the anonymous GitHub platform**, as several users have reported similar cases:
>
> - https://github.com/tdurieux/anonymous_github/issues/395
> - https://github.com/tdurieux/anonymous_github/issues/404
>
> These issues seem related to platform caching or synchronization. At the moment, our anonymous code link (https://anonymous.4open.science/r/MemGAS-626C/README.md) is accessible and the file contents can be viewed normally. If the problem occurs again, please feel free to reach out to me.
>
>
> --------------
>
>
> > ### `Q1-(1):` Which specific stage does the "Avg. Tokens" count refer to?
>
> We sincerely thank the reviewer for pointing out this issue. We clarify that **Avg. Tokens measures the tokens consumed during the *online QA stage* for each  query**. Formally:
> $$
> \text{Avg. Tokens per query} = \text{Tokens used for LLM Filtering} + \text{Tokens used for QA Generation}.
> $$
>
> **Note:**
> - It does NOT include any LLM tokens used for memory construction. It reflects **inference-time token usage per query**, which allows a fair, per-query comparison across different methods.
>
> ---
>
> > ### `Q1-(2):` How is this "Avg. Tokens" measure normalized to be directly comparable to the token counts of single-granularity baselines that may not utilize LLM filtering? Can the authors provide an analysis on token consumption breakdown?
>
> We sincerely thank the reviewer for raising this constructive question. To address the reviewer’s concern about fairness and comparability, we provide a detailed breakdown of `memory construction cost` and `online per-query cost` across different methods. **This experiment has been added in `Section 3.4` in the revised version.**
>
> **(1) Memory construction and online token cost breakdown**
>
> We compare MemGAS with representative baselines in terms of **LLM token usage for memory construction** and **average per-query token cost** (with and without filtering), along with the corresponding GPT4o-J scores (LongMemEval-s):
>
> (Comparison with representative baselines in token cost and QA performance (GPT4o-J Score).)
> | **Category**                     | **setting**                               | **MemGAS (Ours)** | **HippoRAG2** | **RAPTOR** | **SeCom** | **RecurSum** |
> |----------------------------------|-------------------------------------------|------------------:|--------------:|-----------:|----------:|-------------:|
> | **Memory construction**          | Input tokens                              | 52.9M (100%)     | 111.1M (210%) | 62.6M (118%) | 106.2M (200%) | 58.3M (110%) |
> | **Memory construction**          | Output tokens                             | 5.2M (100%)      | 10.9M (210%)  | 0.73M (14%) | 71.1M (1367%) | 16.3M (313%) |
> | **Avg. Tokens per query**        | With filter                               | 8,829            | 8,911         | 6,617      | 3,015      | 9,176        |
> | **Avg. Tokens per query**        | Without filter                            | 8,481            | 8,530         | 6,254      | 2,741      | 8,853        |
> | **QA performance**               | With filter                               | **60.2**         | 58.4          | 33.2       | 56.6       | 36.2         |
> | **QA performance**               | Without filter                            | **59.4**         | 57.6          | 32.2       | 56.0       | 35.4         |
>
> **Observations:**
> - **Memory construction cost**: MemGAS is  more efficient in memory construction than multi-stage structured methods such as HippoRAG2 and SeCom, both in input and output tokens.
> - Under both with filter and without filter settings, MemGAS has **comparable or lower online token usage** than strong structured baselines, while achieving higher scores. The filtering step introduces only a small additional token overhead, typically around 200–300 tokens per query.
>
>
> **(2) MemGAS Token Cost Breakdown**
> During the memory construction stage, the extra tokens introduced by multi-granularity memory (summaries + keywords) are extremely small relative to the original corpus:
>
> | Component | Total Tokens |
> |----------|--------------|
> | **Original Memory** | 51.6 million |
> | **Summary Memory (Ours)** | 3.14 million |
> | **Keywords Memory (Ours)** | 2.09 million |

---

> ### Author Response · Authors · 2025-11-21
> **Response to Reviewer Lozm [Part 4]**
>
> > ### `Q2:` The importance of each granularity could vary for each memory bank $M_i$. Why does Equation 4 assign an overall weight to each granularity for the entire memory bank instead of each $M_i$?
>
> Thank you for raising this thoughtful and insightful question. **The key reason is that our entropy-based router is designed to capture the *global relevance distribution* of each granularity with respect to the query**, which fundamentally relies on the *entire memory bank* rather than any individual memory unit $M_i$.
>
> `Equation 4` computes the entropy $H^g$ using the normalized similarity distribution $p_g$ across all memory chunks at granularity $g$. This entropy reflects how confidently or ambiguously the query matches that granularity overall. However, **an individual memory unit $M_i$ does not contain enough information to estimate this distribution**, nor can it model the uncertainty of the query–granularity alignment in a statistically meaningful way. Because entropy intrinsically measures distribution-level uncertainty, it must be derived at the level of the entire memory bank.
>
> We appreciate your observation, and we hope this explanation clarifies the rationale behind our design.
>
>
>
>
>
> -----
>
> > We hope these responses address your concerns comprehensively. Feel free to let us know if you have any additional questions, and we are more than happy to provide further clarification on any aspect of our work.
> >
> > Respectfully and sincerely,
> >
> > **Your Authors**

---

> > ### Comment · Reviewer_Lozm · 2025-11-25
> >
> > I appreciate the authors for their detailed response and strong results in the additional experiments. I only have one further concern about the response:
> > W1: Although strong experimental results are provided in the response to W1-(2), an ablation study that cuts all cross-granularity edges can be more straightforward to support the claim "our improvements do not come from simply using these granularities, but from our multi-granularity memory framework itself".

---

> > > ### Author Response · Authors · 2025-11-26
> > > **Response to Reviewer Lozm**
> > >
> > > **Dear Reviewer Lozm:**
> > >
> > >
> > > Thank you for your follow-up comment! We sincerely appreciate your thoughtful suggestion and the time you took to carefully review our previous response. Following your advice, we conduct additional ablation study:
> > >
> > > - **MemGAS (cuts all cross-granularity edges) Setup**: We remove all cross-granularity edges that connect nodes from different granularities, so that there is no relevance link across granularities. This operation disables our proposed multi-granularity memory association mechanism and disables the PPR–based retrieval, while only retaining the soft router weights over granularities.
> > >
> > >
> > > | Method                                       | Recall@3 | Recall@5 | Recall@10 |
> > > |----------------------------------------------|:--------:|:--------:|:---------:|
> > > | HippoRAG2                                    | 75.53    | 84.68  | 91.28     |
> > > | MemGAS (w/ all-granularity)                  | **78.51** | **88.94** | **94.47** |
> > > | MemGAS (cuts all cross-granularity edges)    | 75.89    | 84.10    | 90.49     |
> > >
> > > **Findings.**
> > > - **Removing all cross-granularity edges leads to a significant performance drop**, demonstrating that our memory association method is essential for strong retrieval.
> > > - We further observe the drop is especially large at Recall@5 and Recall@10, where HippoRAG2 also performs better. This indicates that **our cross-granularity association mechanism and structured approach (HippoRAG2’s knowledge graph) are better to capture strong contextual relevance memories by constructing edges**, rather than focusing solely on the top few high-similarity candidates (e.g., the top-3).
> > >
> > >
> > > Thank you again for your constructive comments. We hope the response address your concerns.  If you have any further questions, we would be more than happy to clarify them.
> > >
> > > Respectfully and sincerely,
> > >
> > > **The Authors**

---

> > > > ### Comment · Reviewer_Lozm · 2025-11-26
> > > >
> > > > I thank the authors for their extensive experiments and detailed response. The results effectively demonstrate the validity of the proposed mechanism. I will raise my score to 6.

---

> ### Author Response · Authors · 2025-11-26
> **Warm Appreciation to Reviewer Lozm**
>
> **Dear Reviewer Lozm**:
>
> We sincerely thank you for raising the score from **4 to 6**! Your insightful and constructive feedback has greatly improved our paper.
>
> Respectfully and sincerely,
>
> The Authors

---

### Author Response · Authors · 2025-11-29
**Rebuttal Summary**

**Dear Reviewers, AC, SAC, and PC,**

We sincerely thank you for your time, efforts, and careful evaluation of our manuscript. We are also grateful that all reviewers actively engaged in the rebuttal discussion: they indicated that **their main concerns had been addressed and expressed an overall positive view of the work**. We greatly appreciate the reviewers’ constructive comments, which have substantially improved the quality of our work.

For convenience, we summarize below the reviewers’ comments in a table, followed by the key strengths highlighted in the reviews and the key revisions we made.

---

| Reviewer | Rating (Before → After) | Reviewers' Comments After Rebuttal|
|----------|---------|-------|
| Lozm     | **`4 → 6`**| "The results effectively demonstrate the validity of the proposed mechanism. I will raise my score to 6." |
| gP9f     | 6 → 6                   | "Most of my concerns regarding the method's ambiguity and experimental details have been addressed.... so I will keep the positive score as is." |
| Bona     | 8 → 8| "Thanks for the great rebuttal and work from authors, all my concerns have been solved. I will maintain my initial score." |
| tvD3     | **`4 → 6`**| "I appreciate the authors’ rebuttal. The response is helpful, I decide to raise my initial score." |
----

(* We refer to Reviewer Lozm as R1, Reviewer gP9f as R2, Reviewer Bona as R3, Reviewer tvD3 as R4.)



**Key strengths highlighted by the reviewers:**

- **S1: Strong novelty and clear motivation.** Reviewers agreed that our work introduces a novel multi-granularity long-term memory framework that addresses the limitations of single/fixed-granularity memory, with a clear motivation around the information–noise trade-off. (`R1, R2, R3, R4`)
- **S2: Well-structured and convincing module design.** Reviewers considered our GMM-based memory association with graph/PPR-based retrieval—to be well structured and convincing, providing interpretability while remaining concise and flexible. (`R1 R2 R4`)
- **S3: Strong empirical performance.** Reviewers highlighted solid and comprehensive experimental results, with consistent gains over strong baselines on multiple benchmarks for both QA and retrieval. (`R1, R2, R3, R4`)
- **S4: Impressive effectiveness and efficiency trade-off.** Reviewers emphasized the impressive trade-off between performance and latency, noting that our method improves accuracy while keeping token usage and latency competitive with or better than baselines. (`R1, R2, R3, R4`)
- **S5: Clear presentation and supporting analyses.** Reviewers found the paper well-motivated, easy to follow, and elegantly presented, with fine-grained analyses that clarify the contribution of each component. (`R1, R3, R4`)

---

**Key revisions in response to the comments:**
- **[R1, R4] Comparison with hierarchical and compressed-memory methods**. To address the request for more baselines, we added H-MEM and COMEDY baselines as suggested. We included brief descriptions in `Sec. 1` and detailed setup/results in `Appendix I.2`, and found that our method consistently outperforms both baselines.
- **[R1, R2, R3] Clearer explanations of settings and method.** To resolve ambiguities, we added detailed definitions of all query types in `Appendix A`, refined the method description in `Sec. 2.2`,  clarified the retrieval hyperparameters in `Sec. 3.1`, and explained granularities settings in `Appendix C.2 and C.3`, as well as the setup of error analyses in `Appendix G`.
- **[R1, R4] More time and token cost analysis.** To strengthen fairness and practicality claims, we compared our method with representative baselines on memory-construction token cost and average per-query tokens, added a latency breakdown demonstrating that the cost of our modules is relatively small, in `Sec. 3.4` and `Appendix D`.
- **[R2, R4] Human evaluation and reliability of LLM judges.** To address concerns about the reliability of LLM-as-Judge, we conducted human evaluation, reported in `Appendix I.1`, showing strong alignment and low variance.
- **[R4] Scaling experiments.** To address questions about scalability as the memory store grows, we conducted data-scaling experiments from 20k → 1M memory tokens, measuring GMM update latency, PPR convergence time, and LLM construction token usage, and we summarized the patterns and added scaling curves in `Appendix I.3`.
- **[R1] Role and ablation of the LLM filter module.** To make the impact of the LLM information filter explicit, we added a dedicated filter ablation in `Sec. 3.4`, showing that PPR mainly improves retrieval relevance while the filter improves downstream QA by removing noise.

All revisions have been highlighted in ${\color{blue}\text{blue}}$.

------

Thank you again for your time, thoughtful evaluation. We truly appreciate your efforts and hope this summary is helpful.

Respectfully and sincerely,

**The Authors**

---

### Meta-Review · Area_Chair_tkBC · 2026-01-05

**Summary:**

This work studies long-term memory for conversational agents under limited context windows. The authors propose MemGAS, a training-free framework that (i) represents memory at multiple granularities (session / turn / summary / keyword), (ii) builds a multi-granularity association graph via GMM-based accept/reject linking, (iii) uses an entropy-based router to adaptively weight granularities per query, (iv) retrieves with PPR, and (v) applies an LLM filter to reduce redundancy/noise before generation. Across four benchmarks, the method reports improvements on QA and retrieval metrics with competitive token/latency trade-offs.
Post-rebuttal, the paper has strong empirical support, substantially improved clarity, and convincing validation of its core multi-granularity association + adaptive selection mechanism.

**Reviewer Concerns:**

1. Missing some baselines, such as H-MEM and COMEDY (tvD3, Lozm).

2. The definition of token-level metrics is unclear (Lozm, gP9f).

3. The effectiveness of the method is questioned, including granularity choices, modeling stability, and the rationality of the entropy-based router design (tvD3, Lozm, gP9f).

**Reviewer Scores:**

Reviewer Lozm is likely to raise the score to 6, as the authors added hierarchical baselines, clarified token accounting, and provided filter ablations during the rebuttal. In particular, the cross-granularity boundary-cutting ablation directly validates the core mechanism.

Reviewers Bona and gP9f are likely to maintain their scores. The authors addressed ambiguities and experimental clarity in the rebuttal and added more in-depth analyses of efficiency and individual modules.

Reviewer tvD3 is likely to raise the score to 6, as the authors added extensive new experiments during the rebuttal, including robustness studies (encoder/generator), sensitivity and scaling analyses, additional baselines, clearer evaluation protocols, and adaptive-
K experiments.

---

### Decision · Program_Chairs · 2026-01-26

Accept (Poster)